# Illuminating the dark space of neutral glycosphingolipidome by selective enrichment and profiling at multi-structural levels

Zidan Wang [1], Donghui Zhang [2], Junhan Wu [2], Wenpeng Zhang[2] & Yu Xia [1] ✉

Glycosphingolipids (GSLs) are essential components of cell membranes, particularly enriched in the nervous system. Altered molecular distributions of GSLs are increasingly associated with human diseases, emphasizing the significance of lipidomic profiling. Traditional GSL analysis methods are hampered by matrix effect from phospholipids and the difficulty in distinguishing structural isomers. Herein, we introduce a highly sensitive workflow that harnesses magnetic $TiO_2$ nanoparticle-based selective enrichment, charge-tagging Paternò–Büchi reaction, and liquid chromatography-tandem mass spectrometry. This approach enables mapping over 300 distinct GSLs in brain tissues by defining sugar types, long chain bases, N-acyl chains, and the locations of desaturation and hydroxylation. Relative quantitation of GSLs across multiple structural levels provides evidence of dysregulated gene and protein expressions of FA2H and CerS2 in human glioma tissue. Based on the structural features of GSLs, our method accurately differentiates human glioma with/without isocitrate dehydrogenase genetic mutation, and normal brain tissue.

Glycosphingolipids (GSLs) belong to a class of amphipathic and low-abundance lipids that can be ubiquitously found in plasma membrane. They mainly serve as a key structural component of lipid raft microdomains which facilitate membrane trafficking and signal transduction[1]. GSLs are assembled from three building blocks, the glycan moiety (headgroup), long-chain base (LCB), and acyl chain (representative structures shown in Fig. 1a). The biosynthesis of GSLs is regulated by a variety of enzymes (Fig. 1b)[2], which ultimately generates a large diversity of chemical structures, including many isomers.

Besides the well-studied structural variants of the sugar moieties, the fatty acyls in GSLs comprise a plethora of structures. The chain length may vary from C14 to C36; the C2 position can be OH modified by fatty acid 2-hydroxylase (FA2H)[3]; a carbon-carbon bond in the acyl chain can be converted to a double bond (C = C) by desaturases at a specific location[4,5]. Four types of LCBs, including dihydrosphingosine (DHS), sphingosine (SPH)[6,7], phytosphingosine (PHS), and sphingadiene (SPD)[8], are available to be assembled onto fatty acyls under the regulation of ceramide synthases 1-6 (CerS1-6)[9]. Any disturbance to the biosynthetic pathways leads to altered molecular profile of GSLs, which is increasingly linked to neurological diseases[10], metabolic disorders[11,12] and cancers[13]. For instance, GSLs consisting of C16 sphingosine as the LCB were found elevated in the human plasma of Type II diabetes[14], while those having C22-C24 fatty acyl chains were found decreased in the brain of aged mouse[10].

[1]MOE Key Laboratory of Bioorganic Phosphorus Chemistry & Chemical Biology, Department of Chemistry, Tsinghua University, Beijing 100084, China. [2]State Key Laboratory of Precision Measurement Technology and Instruments, Tsinghua University, Department of Precision Instrument, Beijing 100084, China. ✉e-mail: xiayu@mail.tsinghua.edu.cn

**Fig. 1 | Strutural diveristy and biosynthesis of glycosphingolipids. a** Representative structure of a glycosphingolipid and the three building blocks. **b** A simplified scheme of the biosynthesis of mammalian glycosphingolipids.

SCD1 Stearoyl-CoA Desaturase-1, FA2H fatty acid 2-hydroxylase, FADS3 fatty acid desaturase 3, DEGS1-2 dihydroceramide desaturase 1 and 2, UDP glycosyltransferase.

Clearly, profiling of GSLs with detailed structural information can provide highly relevant and useful information for the discovery of altered GSL metabolism due to disease at the molecular level. However, hampered by the structural complexity and low abundance of GSLs in mammalian lipidome (e.g., HexCer <3 mol% of total lipids in the brain tissue), profiling of GSLs, especially neutral GSLs, stands as a long-time challenge even by employing the state-of-the-art lipidomic analysis workflows[15]. For instance, Horˇejší et al. identified 59 structures of neutral GSLs at the chain composition level from human plasma by hydrophilic interaction liquid chromatography (HILIC)-tandem mass spectrometry (MS/MS)[16]. Baba et al. characterized 22 structures of GSLs at the C = C location level from porcine brain lipid extract via electron impact excitation of ions from organics (EIEIO)-MS/MS[17] via shotgun analysis. These numbers of identification however are far smaller than the neutral GSL structures (>2300) curated in the structural database of LIPID MAPS[18].

Considering that phospholipids are ~20 times more abundant than neutral GSLs in brain tissue[19] which strongly suppress the detection of GSL by mass spectrometry, we believe one key step to boost the detection for GSL is to develop a method that can effectively remove phospholipids while selectively enrich neutral GSLs in biological samples. Selective enrichment is commonly employed before MS analysis to improve detection for low abundance constituents, such as glycoproteins and phosphoproteins, in proteomics studies[20]. As compared to micro-columns[21], dispersed microspheres[22], or pipette tips[23], functional magnetic nanoparticles exhibit attractive features for selective enrichment due to their large surface area, flexible surface modification capabilities, and strong magnetic responsiveness[24,25]. Furthermore, MS/MS methods which can resolve structural isomers should be incorporated to address the issue of structural complexity of GSLs. Given these considerations, we have developed a selective enrichment method employing TiO₂ magnetic nanoparticles (TiO₂ MNPs), which achieves ~30-fold enrichment of neutral GSLs from lipid extracts with high recovery rates and reproducibility. This enrichment method is further streamlined with offline C = C derivatization via charge-tagging Paternò–Büchi (PB) reaction[26,27], and reversed-phase liquid chromatography-tandem mass spectrometry (RPLC-MS/MS). The analytical power of this workflow is demonstrated by profiling more than 300 GSL species from porcine brain at detailed structural level across four orders of magnitude in relative abundances. This workflow further enables the discovery of altered GSL metabolism in human glioma tissue samples and the phenotyping of gliomas with isocitrate dehydrogenase genetic mutations (IDH-mutant), gliomas

without IDH mutations (IDH-wildtype), and normal brain tissue samples.

## Results

### Selective enrichment of GSLs by TiO₂ MNPs

Ti (IV) in TiO₂ exhibits strong coordination with cis diol across a broad pH range (1.5-14)[25]. This chemistry has been harnessed for separations of glycosphingolipids[26,27], glycopeptides[28], and nucleosides[29] from complex matrices. Considering that the cis-diol-Ti (IV) coordination is stronger than Ti (IV)-phosphoric acid coordination under basic solution conditions[28], we envision that selective capture of neutral GSLs can be potentially achieved using TiO₂ surfaces by manipulating the pH values in the capture, washing, and release steps. On the other hand, magnetic nanoparticles have analytical advantages of good dispersity in solution, large surface area, and the ease of separation[25]. We thus employed TiO₂ MNPs with a core-shell structure for the development of a selective enrichment procedure for neutral GSLs.

The TiO₂ MNPs were synthesized following an established procedure (Supplementary Fig. 1a)[29,30]. On average, the particles were $305 \pm 22$ nm in diameter, measured from 105 TiO₂ MNPs in a scanning electron microscopy (SEM) image (Supplementary Fig. 1b, c). The ferrite core was 250 nm in diameter and covered by a mesoporous TiO₂ layer of 40 nm thickness (Supplementary Fig. 1d-f). Energy dispersive X-ray (EDX) and X-ray photoelectron spectroscopy (XPS) measurements of the TiO₂ MNPs proved a composite of iron oxide and titanium (IV) dioxide (Supplementary Fig. 1g-i). The TiO₂ MNPs had an average surface-to-volume ratio of 89.9 m² g⁻¹, pore volume of 0.11 cm³ g⁻¹, surface charge of −0.493 mV in neutral distilled water, and good magnetic responsiveness of 44.6 emu g⁻¹ (Supplementary Fig. 2).

A series of mixtures of lipid standards consisting of five types of neutral GSLs and one each representing phosphatidylcholine (PC), phosphatidylethanolamine (PE), and sphingomyelin (SM) were prepared for the selective enrichment tests. The concentrations of PC, PE, and SM standards (40 μg in total) were 20–10,000 times higher than that of GSLs (0.008-2 μg for each standard), mimicking the concentrations of lipids in the brain lipidome[19]. Three different buffer solutions were used for the loading, washing, and eluting steps, respectively. The loading buffer was optimized to contain 6% aqueous NH₃ and 94% acetonitrile (ACN) with a measured pH of 11.83. TiO₂ MNPs (5 mg) was blended with 400 μL of loading buffer in the capture step (Fig. 2a). After 1-hour stirring, the TiO₂ MNPs were washed twice by a washing buffer (4% aqueous NH₃/96% MeOH, v/v, 20 mM NH₄HCO₃, pH=10.18, 400 μL each wash) to remove phospholipids.

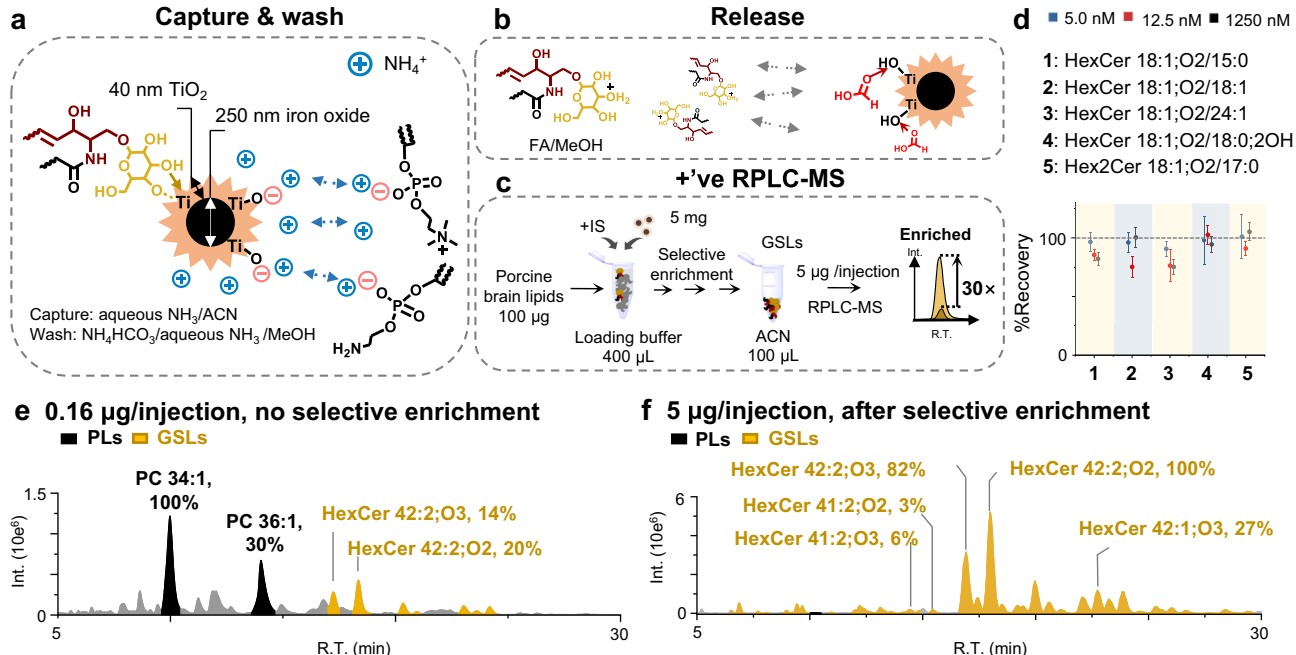

**Fig. 2 | Selective enrichment of GSLs by TiO₂ MNPs. a** Capture of GSLs on TiO₂ MNPs and the removal of phospholipids using alkaline buffer, **b** release of GSLs using acidic buffer, and **c** subsequent analysis by RPLC-MS in positive ion mode. **d** %Recovery of five GSL standards (HexCer 18:1;O2/15:0, HexCer 18:1;O2/18:1, HexCer 18:1;O2/24:1, HexCer 18:1;O2/18:0;2OH, Hex2Cer 18:1;O2/17:0) prepared as mixtures with initial concentrations at 5.0 nM, 12.5 nM, 1250 nM, respectively. Data points represent mean values ± standard deviation (SD) of three technical replicates. RPLC-MS analysis of the equivalents of **e** 0.16 μg of porcine brain lipid extract/ injection without selective enrichment and **f** 5 μg/ injection after TiO₂ MNP-based enrichment. Source data is provided as a Source Data file.

The presence of ammonium ions in the washing buffer was found critical to remove phospholipids, which increased the washing efficiency by 3–8 times relative to the buffer without $NH_4HCO_3$ (4% aqueous $NH_3$/96% MeOH, v/v, Supplementary Fig. 3). We hypothesize that the excess of ammonium cations weakens electrostatic interactions between phospholipids and TiO₂ MNPs under basic pH condition (pH = 10.18) and separates them by forming a positive ion layer on the TiO₂ MNPs (Fig. 2a). Indeed, a zeta potential of 43.3 mV was measured for TiO₂ MNPs in the ammonium containing washing buffer (Supplementary Fig. 2c). Meanwhile, neutral GSL molecules, having the pKa slightly above 12[31], should stay in a neutral state and retain strong coordination with TiO₂ MNPs. Finally, GSLs were released from TiO₂ MNPs after a 2-hour agitation in 400 μL of acidic eluting buffer (5% formic acid/95% MeOH, v/v, pH=1.29, Fig. 2b). The released GSLs were dried and reconstituted in ACN before analysis by RPLC-MS (Fig. 2c).

Recovery was evaluated for five GSL standards (HexCer 18:1;O2/ 15:0, HexCer 18:1;O2/18:1, HexCer 18:1;O2/24:1, HexCer 18:1;O2/ 18:0;2OH and Hex2Cer 18:1;O2/17:0) which had different acyl chain lengths (C15-C24), degrees of unsaturation (1-2), number of hydroxyl groups (0-1), and the sugar headgroups (HexCer and Hex2Cer). The GSL standards were prepared at three initial concentrations (5 nM, 12.5 nM, and 1250 nM), then selectively enriched using TiO₂ MNPs, and concentrated four times before MS analysis. %Recovery was assessed by comparing the ion abundance of each GSL (sum of [M + H]⁺, [M + Na]⁺, and [M-H₂O + H]⁺) before and after selective enrichment. The good %recovery was achieved, ranging from 75–105% (Fig. 2d and Supplementary Fig. 4). The detection limit for the neutral GSL was estimated to be 20 nM. The amount of captured GSLs also showed a good linear dynamic range (R² ≥ 0.990) for concentrations spanning three orders of magnitude (Supplementary Fig. 5). Although Ti(IV) can catalyze the hydrolysis of phosphate diesters[32], we did not observe any hydrolysis products of GSLs or phospholipids.

We also evaluated %recovery of lipid standards representative of other subclasses of neutral GSLs and sulfatides, including Hex3Cer 18:1;O2/18:0, Gb4 18:1;O2/24:0, SHexCer 18:1; O2/24:1, SHexCer 18:1;O2/17:0, and SHex2Cer 18:1;O2/18:0. Except for SHexCers (<50%), the %recoveries of the rest all exceeded 70% (Supplementary Fig. 6). We hypothesize that the C3-sulfation disrupts the cis-diol configuration and weakens the coordination of SHexCer with Ti(IV) during the loading step, leading to compromised recovery. Furthermore, the current selective enrichment condition is not optimal for gangliosides, because they can form stable carboxylic acid-Ti(IV) coordination under acidic washing condition and thus cannot be effectively eluted from the surface[33]. Consequently, in later analysis we primarily target neutral GSLs from brain lipid extracts.

The performance of TiO₂ MNPs was further evaluated using polar lipid extract of porcine brain. The lipid extract (100 μg) was mixed with 5 mg of TiO₂ MNPs in the loading buffer (400 μL). The enrichment procedure followed the same steps as those used for standard lipid mixtures. Figures 2e and f compare the base peak chromatogram (BPC) from RPLC-MS¹ ([M + H]⁺) of 0.16 μg lipid extract/injection without selective enrichment to that of 5 μg/injection, equivalent to an increase of 30 folds of lipid quantity after selective enrichment. The interfering phospholipids, such as PC 34:1 and PC 36:1 were removed with high efficiency (>99%, Supplementary Table 1), while the signals of a variety of GSLs become dominant after enrichment (Supplementary Data 1). It should be noted that without applying selective enrichment, an injection of 5 μg lipid extract would overload the column, resulting in poor detection of GSLs. We compared the relative abundances of 22 neutral GSLs consisting of different chain lengths and OH groups in porcine brain before and after selective enrichment. Almost identical profiles were obtained (RSD < 15%, Supplementary Fig. 7). TiO₂ MNP-based enrichment for GSL also provided good reproducibility in parallel experiments. The ion signals of representative GSLs of different subclasses all exhibited less than 20% RSD from five replicates, even though their MS¹ signals spanned three orders of magnitude (Supplementary Fig. 8). TiO₂ MNPs synthesized from different batches exhibited less than 5% variations regarding the ion abundances of two main components of brain GSLs, HexCer 42:2;O2 and HexCer 42:2;O3 (Supplementary Fig. 9a-b). Moreover, the variations were less than

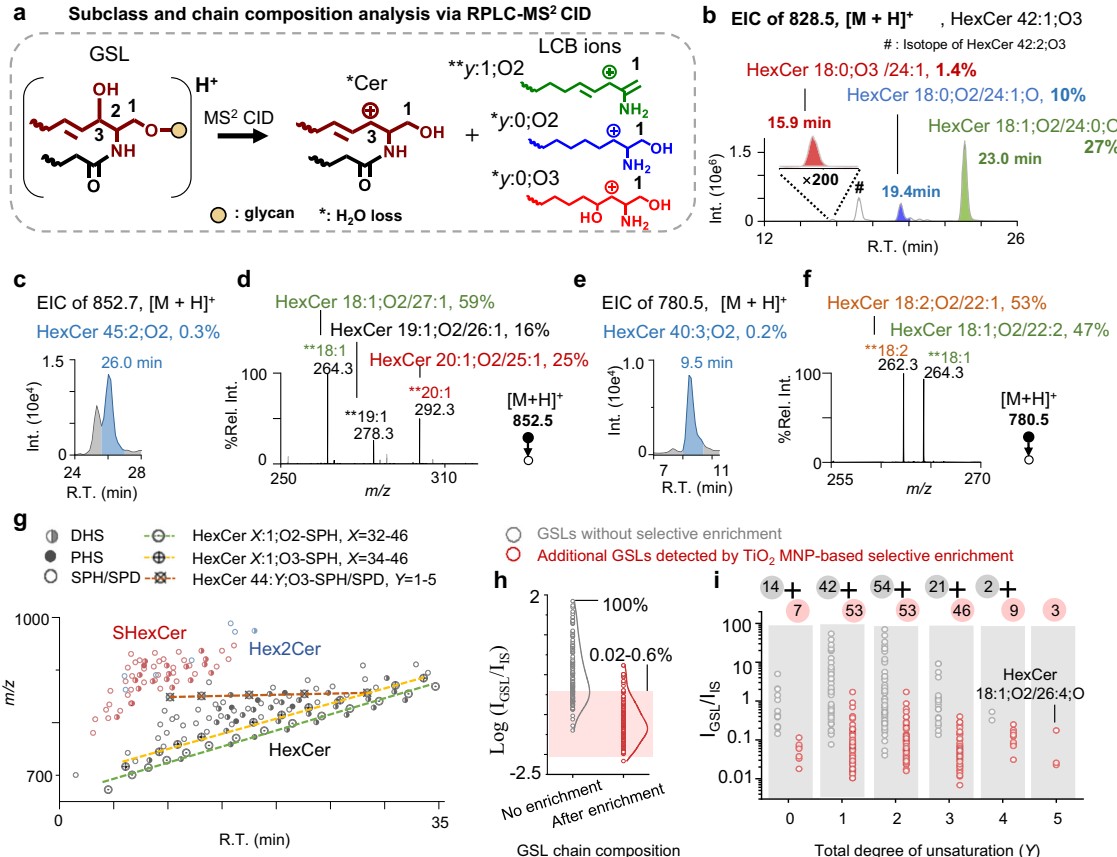

**Fig. 3 | Profiling of GSLs at the class and chain composition levels from porcine brain lipid extract. a** Generic structures of *Cer and LCB fragment ions formed from MS² CID of GSL ([M + H]⁺). **b** Extracted ion chromatogram (EIC) of HexCer 42:1;O3 (*m/z* 828.5, [M + H]⁺). **c** EIC of HexCer 45:1;O2 (*m/z* 852.5, [M + H]⁺) and **d** corresponding MS² CID spectrum showing the region for LCB fragment ions. **e** EIC of HexCer 40:3;O2 ([M + H]⁺, *m/z* 780.5) and **f** corresponding MS² CID spectrum showing the region for LCB fragment ions. **g** A plot of *m/z* ([M + H]⁺) vs. retention time (R.T.) of 166 GSLs at the sum composition level with confirmed LCB types.

Distinct trendlines can be observed for HexCer *X*:1;O2-SPH (*X* = 32-46), HexCer *X*:1;O3-SPH (*X* = 34-46), and HexCer 44:*Y*;O3 (*Y* = 1-5). **h** Relative quantitation of 304 GSLs at the chain composition level with (5 µg lipid/injection) or without selective enrichment (0.16 µg lipid/injection). IS: HexCer 18:1;O2/15:0, 100 pmol/100 µg porcine brain lipids. **i**. GSLs plotted according to the total degree of unsaturation (*Y* = 0 −5). Data points represent the mean values of three technical replicates. Source data is provided as a Source Data file.

17% after a 3-month storage (Supplementary Fig. 9b-c, Supplementary Table 1).

## Profiling of GSLs at the class and chain composition levels

The success of selective enrichment of GSLs allowed us to profile GSLs with more structural information via RPLC-MS² CID in positive ion mode. We detected three subclasses of GSLs from porcine brain based on observing distinct neutral losses of the sugar moieties, including HexCer (−180 Da), SHexCer (−260 Da) and Hex2Cer (−342 Da) (Supplementary Fig. 10a-c). MS² CID also produced characteristic fragment ions of the LCB, symbolized by *y*:0;O2, **y*:1;O2, and *y*:0;O3, with * denoting H₂O loss and *y* denoting carbon number in LCB (structures shown in Fig. 3a)³⁴. Figure 3b shows the extracted ion chromatogram (EIC) of HexCer 42:1;O3 ([M + H]⁺, *m/z* 828.5). The presence of three distinct EIC peaks suggests isomeric species likely containing different types of LCBs. Indeed, MS² CID of each peak produced characteristic Hex loss ions, while distinct LCB- fragment ions were observed, i.e., *18:0;O3 (PHS), *18:0;O2 (DHS), and **18:1;O2 (SPH) for the peaks eluted at 15.9, 19.4, and 23.0 min, respectively (Supplementary Fig. 11a-c). The isomers were thus confidently identified as HexCer 18:0;O3/24:1, HexCer 18:0;O2/24:1;O, and HexCer 18:1;O2/24:0;O, respectively.

RPLC-MS² CID also allowed analysis of GSL isomers of the same LCB type, but only differing in the combination of the carbon numbers in LCB and N-acyl. HexCer 45:2;O2 (*m/z* 852.5, [M + H]⁺, 26 min, Fig. 3c) is a low abundance lipid, accounting 0.3% of HexCer 42:2;O2 (100%,

Fig. 2f). MS² CID (Fig. 3d) reveals that there are three SPH ions corresponding to different chain lengths, i.e., **18:1;O2 (*m/z* 264.2688, -0.8 ppm), **19:1;O2 (*m/z* 278.2839, 1.4 ppm), and **20:1;O2 (*m/z* 292.2996, 1.0 ppm). Consequently, they are identified as HexCer 18:1;O2/27:1, HexCer 19:1;O2/26:1 and HexCer 20:1;O2/25:1. Additionally, using the relative abundance of **y*:1;O2 ions, relative compositions of the three species within the formula of HexCer 45:2;O2 are determined to be 59%, 16%, and 25%, respectively.

GSL isomers containing LCBs differing in the degrees of unsaturation were also identified. Figure 3e shows that although EIC of HexCer 40:3;O2 ([M + H]⁺, *m/z* 780.5, 9.5 min) only shows one major elution peak (0.2% relative to HexCer 42:2;O2), MS² CID reveals two distinct fragment ions of the LCB, viz., **18:1;O2 (*m/z* 264.2689, -0.8 ppm) and **18:2;O2 (*m/z* 262.2532, 1.1 ppm), which can be attributed to HexCer 18:1;O2/22:2 and HexCer 18:2;O2/22:1, respectively (Fig. 3f). Based on the abundance of the **y*:1;O2 and **y*:2;O2 ions, the relative compositions of the two isomers are determined to be 47% and 53%, respectively.

Figure 3g plots the *m/z* of GSLs at the sum composition level with confirmed LCB types against retention time (R.T.). The three subclasses, HexCer, SHexCer, and Hex2Cer, show distinct retention behavior according to the hydrophobicity of the sugar moieties. For GSLs of the same sum composition, e.g., HexCer 42:1;O3, but containing different types of LCBs, the elution always follows the order of PHS, DHS, and SPH/SPD (co-eluted). Within the same type of LCB,

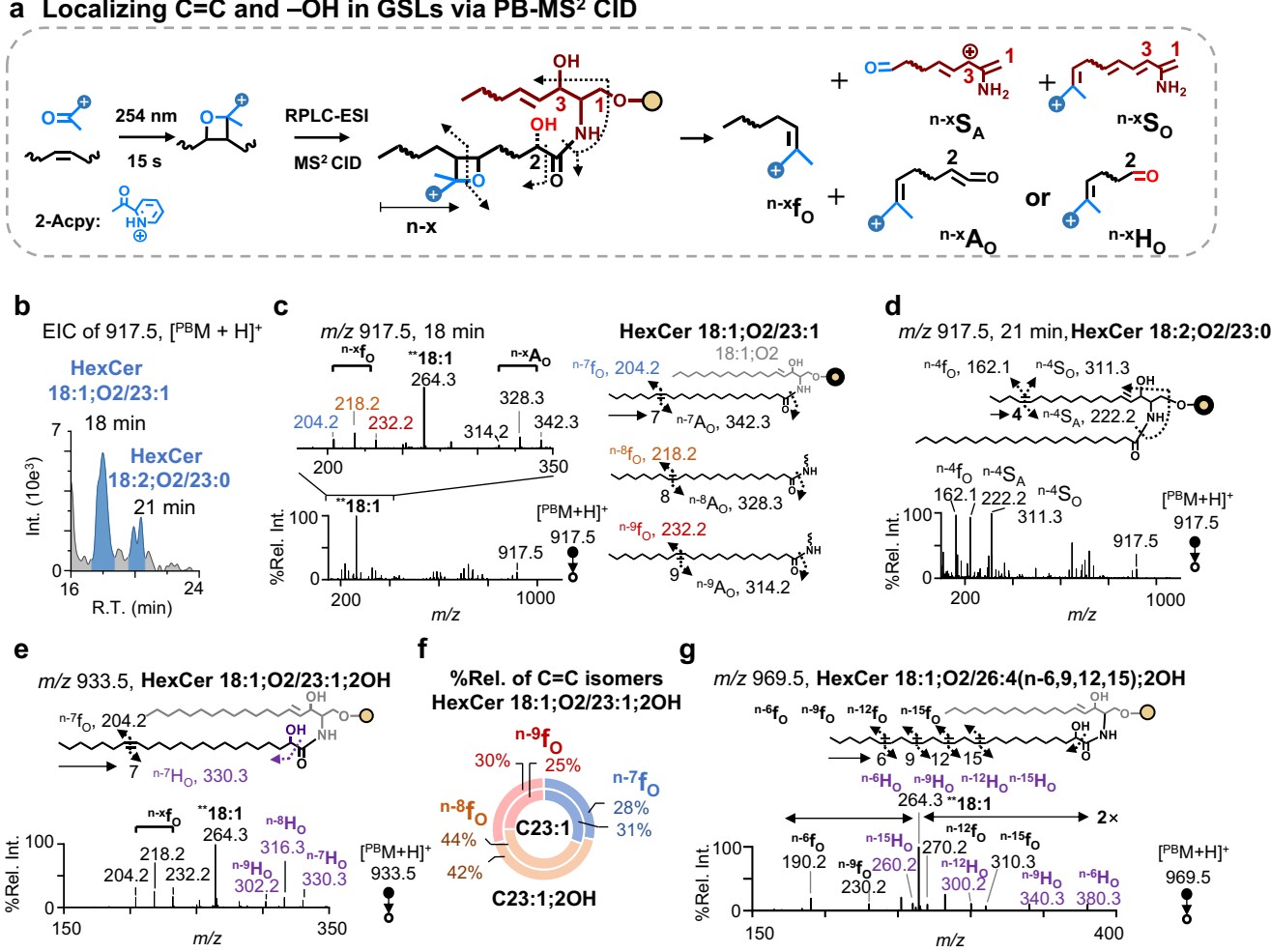

**Fig. 4 | Profiling of porcine brain GSLs at the C = C location level. a** Generic structures of C = C diagnostic ions generated from charge-tagging PB-MS² CID. **b** EIC of HexCer 18:1;O2/23:1 and HexCer 18:2;O2/23:0 derivatized by 2-acpy ([PBM + H]⁺, m/z 917.5). MS² CID spectra for peaks eluted at **c** 18 min and **d** 21 min in panel **b**. **e** MS² CID spectrum of HexCer 18:1;O2/23:1;2OH derivatized by 2-acpy

([PBM + H]⁺, m/z 933.5). **f** %Relative composition of C = C location isomers detected in HexCer 18:1;O2/23:1 and HexCer 18:1;O2/23:1;2OH. **g** MS² CID spectrum of HexCer 18:1;O2/26:4, 2OH derivatized by 2-acpy ([PBM + H]⁺, m/z 969.5). Source data are provided in the Source Data file.

adding one degree of unsaturation effectively shortens the R.T. by 4.5 min (Supplementary Data 2). In addition, HexCer X:1;O3 and Hex-Cer X:1;O2 form distinct trend lines with HexCer X:1;O3 eluting earlier than the latter one (X representing the carbon number). These separations ensure accurate and reliable identification of the low-abundance GSLs, including those with odd-numbered, ultra-long (>26 carbon), and short (<18 carbon), and polyunsaturated N-acyl chains.

Overall, selective enrichment significantly enhances the coverage of GSLs, resulting in an identification of 304 GSL structures at the chain composition level as compared with 133 GSLs without enrichment (Supplementary Data 3). The increase is largely contributed by the detection of medium to low abundance GSLs, i.e., 90% of additionally identified GSL species having relative intensities within the range of 0.02-0.6% relative to HexCer 42:2;O2 (100%) (red shadowed area in Fig. 3h). Furthermore, many low abundance GSLs are found to total degree of unsaturation (Y) higher than 2, suggesting that they contain polyunsaturated chains (Fig. 3i). These GSL structures, which have rarely been reported previously[2], are only detectable after selective enrichment.

### Profiling of GSLs at the C = C location level

The detection of many unsaturated GSLs (Fig. 3i) promoted us to seek profiling of GSLs at the C = C location level. A previously developed

charge-tagging PB reagent, 2-acpy (structure shown in Fig. 4a), was employed to derivatize C = C and improve ionization of GSLs at the same time. The offline PB reaction (15 s) and subsequent RPLC-MS² CID conditions were optimized using GSL standards (Supplementary Fig. S12a-e). Even though PB conversion was not quantitative (<50%), the ionization efficiency of derivatized GSL ([PBM + H]⁺) was greatly improved due to charge-tagging effect and reduced in-source fragmentation.

Figure 4a shows the generic structures of C = C diagnostic ions that are generated from PB-MS² CID of unsaturated GSLs. For a C = C located in the N-acyl chain, two types of C = C diagnostic ions are generated for each C = C, i.e., ⁿ⁻ˣf_O and ⁿ⁻ˣA_O. The fragment containing the methyl end is denoted as ⁿ⁻ˣf_O, in which the superscript "n-x" represents the location of C=C following the n-nomenclature, while the subscript "O" signifies an olefin functional group at the cleavage site. Different from the ⁿ⁻ˣf_O, ⁿ⁻ˣA_O carries the carboxylic potion of the acyl chain resulting from sequential cleavages at the amide bond and C = C bond. The occurrence of 2-hydroxylation on the N-acyl chain, referred to as 2OH-FA, leads to a specific cleavage at the 2OH position, forming the ⁿ⁻ˣH_O ion. This fragment type was verified by lipid standard, Cer 18:1(n-14);O2/18:1(n-9);2OH (Supplementary Fig. 12f-h). For the trans n-14 C = C in SPH, very few C = C diagnostic ions can be formed due to steric hindrance from the headgroup[35]. For C = C

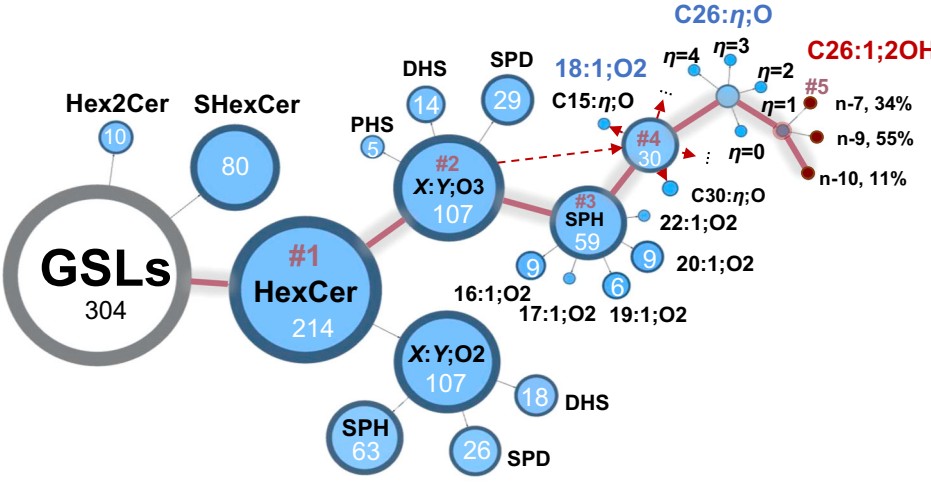

**Fig. 5 | Structural atlas of GSLs in porcine brain.** The number in each circle indicates the count of GSL structures identified in that category, which is directly proportional to the size of the circle. The five-layer hierarchy is denoted by #number. Source data is provided in the Source Data file.

located in the other positions of the LCB, C = C diagnostic ions can be effectively generated, including, $^{n \times}f_O$, $^{n \times}S_A$, and $^{n \times}S_O$, in which the S symbol indicates that the fragment contains the polar portion of the LCB. The limits of detection (LODs) at the C = C location level for HexCer 18:1(n-14);O2/18:1(n-9) and SHexCer 18:1(n-14);O2/18:1(n-9) were 5 nM and 10 nM (Supplementary Fig. 12d-e), respectively. Due to the incorporation of selective enrichment and charge-tagging PB derivatization, the RPLC-MS/MS method allowed us to identify many low-abundance unsaturated GSLs. Using HexCer 41:2;O2 as an example, chain composition analysis via RPLC-MS² CID showed that it contained two isomers, HexCer 18:1;O2/23:1 (91%) and HexCer 18:2;O2/23:0 (9%) (Supplementary Fig. 13a). While the underivatized isomers coeluted in RPLC separation (Supplementary Fig. 13b), 2-acpy derivatized products were separated under the same elution condition (Fig. 4b), leading to confident structural identification at the C = C location level. Figure 4c shows that PB-MS² CID of HexCer 18:1;O2/23:1 (18 min) produces three sets of C = C diagnostic ions: $^{n-7}f_O/^{n-7}A_O$, $^{n-8}f_O/^{n-8}A_O$, $^{n-9}f_O/^{n-9}A_O$, proving the existence of n-7, n-8, and n-9 in the N-acyl C23:1, respectively. For HexCer 18:2;O2/23:0, $^{n-4}S_A$ and $^{n-4}S_O$ specific to 18:2;O2 LCB and its complementary $^{n-4}f_O$ corroborate the identification of SPD as the LCB (Fig. 4d).

Figure 4e shows PB-MS² CID of HexCer 18:1;O2/23:1;O ([$^{PB}$M + H]⁺, m/z 933.5). In addition to a set of $f_O$ ions ($^{n-7/8/9}f_O$), the $H_O$ ions ($^{n-7/8/9}H_O$), which are specifically related to the 2OH on the unsaturated N-acyl chain are detected. Combining information from these two types of C = C diagnostic ions, we can unambiguously identify HexCer 18:1;O2/23:1;O as a mixture of n-7, n-8, and n-9 C = C location isomers of N-acyl C23:1;2OH. The capability of locating 2OH is a great advantage because it is challenging to identify by conventional MS/MS methods[34,36,37].

Because of the lack of authentic standards for unsaturated GSLs, the relative composition of the C = C location isomers was calculated by normalizing the relative ion abundance of a specific $^{n \times}f_O$ to the summed $f_O$ ions from all C = C isomers observed in the same PB-MS² CID spectrum. Figure 4f compares the %relative composition (%Rel.) of the C = C location isomers in HexCer 18:1;O2/23:1 and HexCer 18:1;O2/23:1;2OH. It is interesting that the compositions are very similar, with n-8 to be the major component followed by n-7 and n-9 isomers.

In addition to mono-unsaturated N-acyls, PB-MS/MS was able to provide high quality data for GSLs consisting of polyunsaturated N-acyls, which were at low abundances in porcine brain. As an example, HexCer 44:5;O3 ([M + H]⁺, m/z 848.5) was only detectable after selective enrichment with HexCer 18:1;O2/26:4;O being the major component (Supplementary Fig. 13c). PB-MS/MS further identified the species as HexCer 18:1;O2/26:4 (n-6,9,12,15);2OH (Fig. 4g).

In summary, we identified 225 GSL structures at the C = C level (data provided in Supplementary Data 4). Among them, 68 structures of GSLs were confirmed to consist of SPH (dy:1, n-14) as the LCB, according to retention time behavior and MS² CID pattern[38]. Additionally, we found that n-7, n-8, n-9, n-10, and n-12 isomers were common components for mono-unsaturated N-acyl chains. For N-acyl chains of two and three C = C bonds, we found n-(6, 9), n-(7,10), and n-(9, 12), n-(6, 9, 12), n-(7, 10, 13), and n-(9, 12, 15) as the major isomers, respectively. For N-acyls consisting of four C = C bonds, only n-(6, 9, 12, 15) was detected. All unsaturated N-acyl chains containing hydroxylation were identified as 2OH-FA; LCB 18:2;O2 existed only as 18:2(n-4, 14);O2.

## The atlas of GSLs in porcine brain

Facilitated by $TiO_2$ MNP-based selective enrichment, charge-tagging PB derivatization, and RPLC-MS/MS, we were able to compile an atlas covering more than 300 distinct structures of GSLs in porcine brain, which have not been achieved before. The atlas is depicted in 5 layers' structural hierarchy (Fig. 5). Starting from the headgroup information, there are three subclasses of GSLs including HexCer, Hex2Cer, and SHexCer. For each subclass of GSLs, they are further classified according to the number of oxygen sites in the ceramide moiety, denoted by O2 or O3 as in HexCer X:Y; O2/O3. HexCer X:Y;O3 contains four types of LCB, including SPH, SPD, DHS, and PHS. The SPH-containing HexCer X:Y;O3 is further categorized into SPH of different chain lengths, SPH y:1;O2, y = 16-20, 22. Within the category of HexCer 18:1;O2/Cx:η;O, the acyl chain lengths (denoted by x) vary from 15 to 30 carbon atoms. Using HexCer 18:1;O2/C26:η;O as an example, the degree of unsaturation in N-acyl (denoted as η) varies from 0 to 4. A deeper look at HexCer 18:1;O2/26:1;O shows three C = C location isomers at n-7 (34%), n-9 (55%), and n-10 (11%), and the OH is located at C-2 on the N-acyl chain. The data in Fig. 5 represent the most comprehensive structure atlas of GSLs to date, revealing a large diversity in the GSL metabolism that have not been uncovered before. It is worth noting that HexCer and Hex2Cer are the two subclasses of neutral GSLs found in porcine brain, while complex neutral GSLs, such as Hex3Cer or Gb4, are not detected. This result is consistent with previous reports utilizing thin layer chromatography separation followed by MS analysis[39], or chiral LC separation paired with MS analysis[40].

## Profiling of GSLs in human glioma tissue at multiple structure levels

Gliomas are the most malignant brain tumors of neuroectodermal origin. Using elevated 2-hydroxyglutarate as the biomarker, better

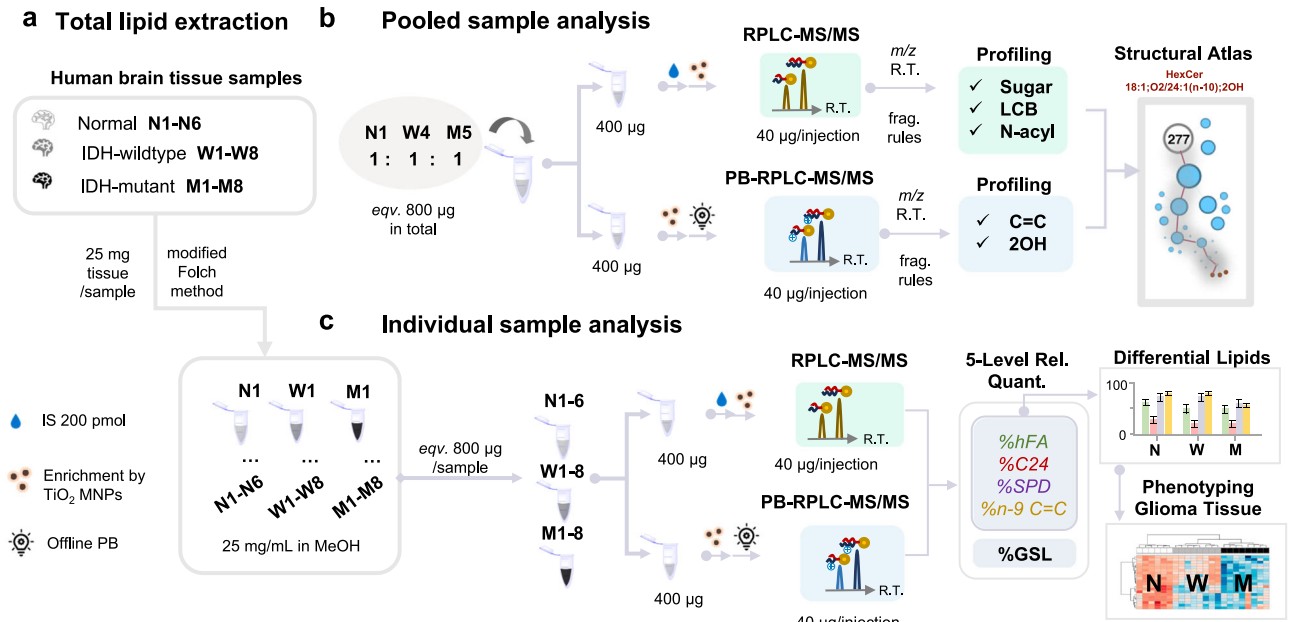

**Fig. 6 | Workflow for deep profiling of GSLs in human glioma tissue samples.**
**a** Total lipid extraction from tissue samples (25 mg/each) following the modified Folch method. **b** Analysis of the pooled sample mixed by 1:1:1 from one normal (N1), one IDH-wildtype (W4), and one IDH-mutant (M5), was performed to build the structural atlas. **c** Analysis of individual samples (Normal, N1-N6, $n = 6$; IDH-wildtype, W1-W8, $n = 8$; IDH-mutant, M1-M8, $n = 8$) was performed to achieve relative quantitation at the sum composition level (%GSL), chain composition level (%hFA, %C24, %SPD), and C = C location level (%n-9), using four-level relative quantitation for glioma phenotyping. IS: HexCer 18:1;O2/15:0.

prognoses have be achieved for glioma with isocitrate dehydrogenase genetic mutation (IDH-mutant)[41]. Glioma without IDH mutations (IDH-wildtype)[42], on the other hand, is a more malignant phenotype, with no genetic mutation nor molecular marker being identified so far. Given the potential importance of GSLs in glioma metabolic remodeling[43–45], we are motivated to realize lipidomic phenotyping of IDH-wildtype, IDH-mutant, and normal human brain tissue by leveraging the capability of profiling GSLs at multiple structural levels.

The atlas of human brain GSLs was established following the workflow depicted in Fig. 6. Firstly, total lipids were individually extracted from 22 samples (N1-N6, M1-M8, W1-W8), following a modified Folch method (Fig. 6a). Then the same amount of lipid extracts from M5, W4, and N1 (each equivalent to 133 µg tissue) were mixed to generate the pooled sample with 200 pmol of HexCer 18:1;O2/15:0 added as IS. The pooled sample was separated into two aliquots and subjected to selective enrichment by TiO$_2$ MNPs. The enriched GSLs were then analyzed by two analysis pipelines: 1) untargeted analysis via RPLC-MS/MS in positive ion mode (2 replicates) to profile GSLs at the headgroup and chain composition levels and 2) offline charge-tagging PB derivatization followed by untargeted analysis via RPLC-MS/MS in positive ion mode (2 replicates) to profile GSLs at the C = C location level, in which process the 2OH position in unsaturated N-acyls was also determined. We thus detected 162 GSL structures at the sum composition level and 277 structures at the chain composition level from three subclasses, including HexCer, SHexCer, and Hex2Cer (Supplementary Data 5) and 236 GSL structures at the C = C location level (Supplementary Data 6). The atlas of human brain GSL is provided in Supplementary Fig. 14-17.

Then individual samples (N1-N6, M1-M8, W1-W8) were analyzed. The ion abundances of IS (HexCer 18:1;O2/15:0, 200 pmol) from these samples exhibited RSD less than 20%, suggesting a relatively stable performance from selective enrichment (Supplementary Fig. 18). Relative quantitation for five structural features was then performed (Fig. 6c). These include 1) %GSL at the sum composition level for species with relative abundance >1% of the most abundant GSL; 2) %hFA, the fraction of a GSL molecule containing an hFA chain relative to

its structure analogue without the OH group; 3) %SPD, the fraction of 18:2;O2 LCB isomer relative to its 18:1;O2 isomer; 4) %C24, the fraction of the chain length isomer C24 relative to C26; 5) %n-9, the fraction of n-9 C = C location isomer relative to all other C = C location isomers.

Figure 7a compares RSDs of relative quantitation for these five structural features. At the sum composition level, 40 GSL species were commonly detected with relative abundances higher than 1% of HexCer 42:2;O3, the most abundant neutral GSL in all samples. They were all associated with large RSD values, ranging from 29% to 160%. Among them, five showed significant decreases (*p* values: 0.0067-0.0418) in IDH-mutant relative to normal. Three showed significant changes (*p* values: 0.0071-0.0221) in IDH-wildtype relative to normal, while no GSL with significant change was found between IDH-mutant and IDH-wildtype. The above results represent a common difficulty encountered by using current lipidomic analysis workflows which only perform relative quantitation at the sum composition level. Because each sum composition consists of an array of structural isomers, it is impossible to untangle metabolic changes contributed by individual species.

Among the 40 GSL species, 30 of them were contributed by 15 pairs of HexCer *X:Y*;O2 and their corresponding hFA analogues (HexCer *X:Y*;O3).Therefore, %hFA was calculated for each pair and the associated RSD values were significantly reduced (5-46%). In fact, relative quantitation at the other detailed structural levels all showed much smaller variations in RSD, e.g., 13–69% for %C24, 6–62% for %SPD, and 0.1–20% for %n-9 (Fig. 7a). Benefiting from the significantly reduced sample-to-sample variations, 35 groups of differential GSLs were discovered from relative quantitation at detailed structural features.

Figure 7b summarizes %hFA for 15 pairs of HexCer *X:Y*, O2/O3 (*X* = 36-44, *Y* = 1–3). For both IDH-mutant and IDH-wildtype samples, %hFA was significantly lower compared with that in the normal tissue samples. However, no significant differences were observed between IDH-mutant and IDH-wildtype. Because FA2H is the enzyme responsible for 2-hydroxylation[3], we conducted quantitative reverse transcription polymerase chain reaction (qRT-PCR) and western blot

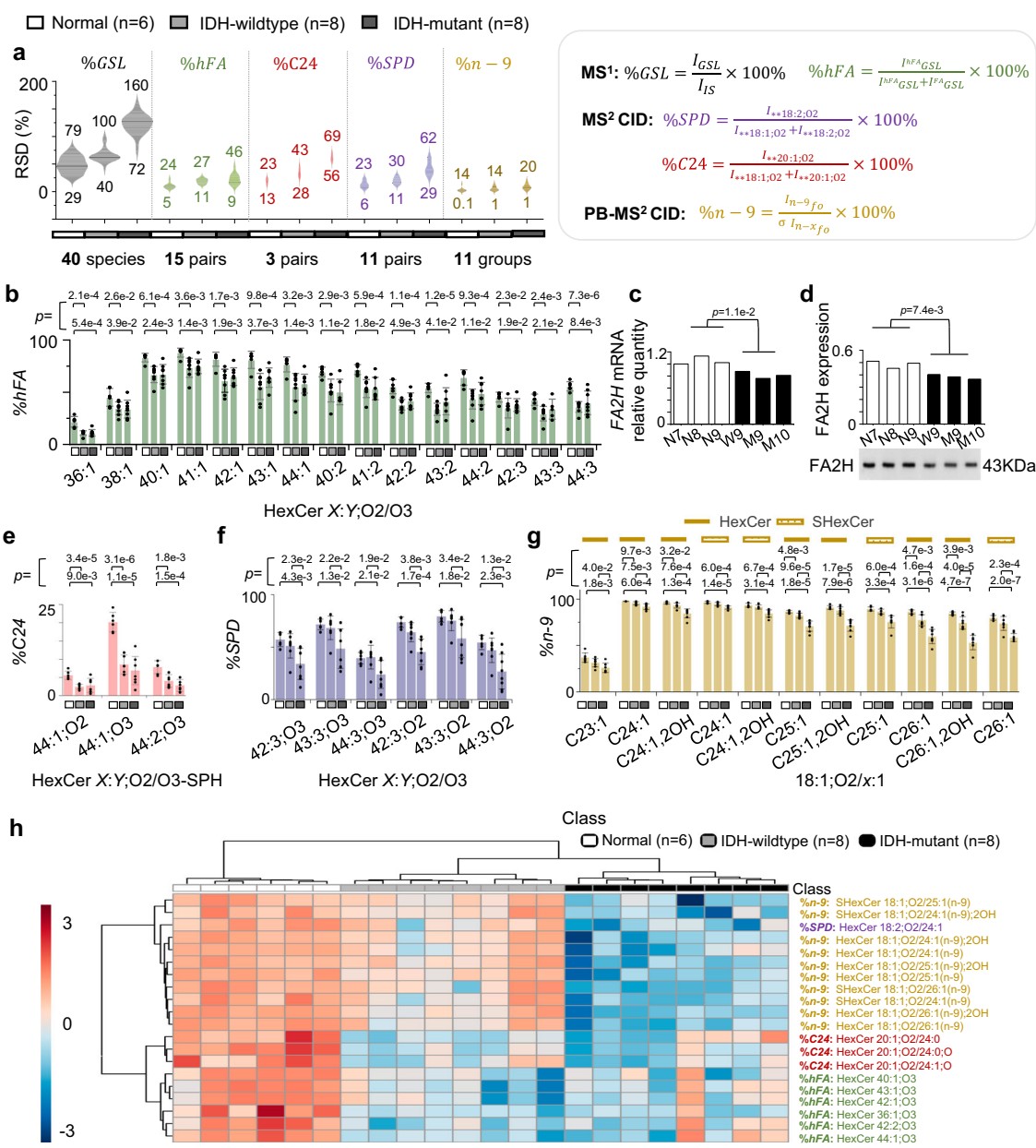

**Fig. 7 | Relative quantitaion of human brain GSLs across multiple structural levels for human glioma phenotyping.** Relative quantitation of GSLs at multiple structural levels from human glioma tissue samples, including normal (N, $n = 6$), IDH-mutant (M, $n = 8$), and IDH-wildtype (W, $n = 8$). A total of 22 biological samples were used in the following figures a, b, e–h. a RSDs for %GSLs from 40 species, %hFA from 15 pairs, %C24 from 3 pairs, %SPD from 11 pairs, and %n-9 C = C isomers from 11 groups. The equations for calculating the values are provided in the inset of panel **a**. **b** Comparisons of %hFA in 15 pairs of GSLs (HexCer $X$:$Y$;O2/O3). **c** Quantitative transcription analysis of FA2H and **d**. Western blot analysis of FA2H from six biological samples: W9, M9-M10, and N7-N9 with one independent

experiment. %C24 of GSLs having SPH as the LCB (HexCer $X$:$Y$;O2/O3-SPH). **f** %SPD of HexCer $X$:$Y$;O2/O3. **g** %n-9 of HexCer and SHexCer consisting of 18:1;O2/$x$:1. **h** Hierarchical cluster analysis using %hFA, %C24, %SPD, and %n-9 for the three sample groups by utilizing the top 20 features with lower $p$ value in t-test. Note: Differences between two groups of samples were all evaluated for statistical significance using the two-tailed Student's t test. Bars represent mean values ± standard deviation (SD). Distance measure: Euclidean, clustering method: Ward. Colors represent % relative amounts/compositions as indicated by the color bar. Data analysis performed by an online software: https://www.metaboanalyst.ca/.Source data are provided as a Source Data file.

analysis for FA2H from 3 normal (N7-N9) and 3 glioma tissue samples (W9, M9, M10). Indeed, in glioma tissue samples the mRNA and the protein level of FA2H were significantly lower (Fig. 7c, d), explaining the metabolic origin of reduced amount of GSLs containing hFA (Supplementary Table 2).

For the three pairs of GSLs containing N-acyl C24 vs. C26 chain isomers, %C24 was consistently lower in glioma tissues as compared to that in the normal tissue as shown Fig. 7e. CerS2 and CerS3 have distinct substrate preferences in assembling acyl-CoAs to the LCB, with

the former one being specific to C24 (Fig. 1a). Indeed, we found that both the transcription and protein levels of CerS2 were significantly lower in glioma tissues as compared to normal control (Supplementary Fig. 19a-b).

The analysis of the 11 pairs of GSL isomers which consist of LCB isomers, viz. SPD vs. SPH showed that 6 pairs had a significant decreased %SPD in the IDH-mutant samples relative to the normal and IDH-wildtype groups (Fig. 7f). At the C = C location level, the compositions of n-9 C = C location isomer (%n-9) in all groups were

significantly lower in IDH-mutant tissue than that of IDH-wildtype and normal samples, while five groups from IDH-wildtype tissues exhibited notable decreases in %n-9 relative to normal tissue (Fig. 7g, Supplementary Table 3). SCD1 is a Δ-9 fatty acid desaturase that catalyzes the synthesis of n-7 and n-9 MUFA[46]. Although the transcription and protein levels of SCD1 were found to be significantly lower in glioma tissues (Supplementary Fig. 19c-d), we could not draw a direct link to decreased %n-9 because the biosynthesis of n-7 MUFAs also utilizes SCD1.

We then conducted unsupervised hierarchical clustering from the 35 groups GSLs which showed significant changes. The top 20 groups ($p < 0.0090$) were able to provide correct clustering among the normal, IDH-wildtype, and IDH-mutant groups (Fig. 7h). The normal group is characterized by an enrichment of all four structural features, viz. % hFA, %SPD, %C24, and %n-9. Differences in %n-9 and %SPD distinguish IDH-wildtype from IDH-mutant, with an enrichment of both features in the IDH-wildtype samples. It's worth noting that neither traditional quantitative methods nor single-level structural features of GSLs can phenotype the three groups correctly (Supplementary Fig. S20, S21). Furthermore, analysis of the total fatty acids released from phospholipids at the C = C location level only discovered one differential lipid ($p$ value: 0.0321) between normal and IDH-mutant tissue and failed to cluster the three groups (Supplementary Fig. S22).

## Discussion

In this work, we demonstrate that the major hurdles for analysis of neutral GSLs can be readily resolved by combining $TiO_2$ MNP-based selective enrichment and isomer-resolved MS/MS techniques. Based on these innovations, we have developed a highly sensitive workflow for profiling of brain GSLs at multiple structure levels, including the sugar head group, LCB and N-acyl composition, 2OH, and C = C location. This method can provide ~30-fold enrichment of GSLs from complex lipid extract and the detection can be as low as 20 pM at the chain composition level. This high sensitivity allows us to depict the structural atlas for more than 300 GSLs in the porcine brain with concentrations spanning four orders of magnitude. Such an extent of lipid coverage represents ~5 times increase than previous reports[47] and ~10 times increase at the C = C location level[17]. Many low-abundance GSLs that could not be seen without selective enrichment are revealed, such as GSL containing poly-unsaturated N-acyls (HexCer 18:1;O2/26:4 (n-6,9,12,15); 2OH). Furthermore, structural analogues or isomers due to difference of OH in LCB vs. N-acyl, LCB being SPD vs. SPH, and different C = C location in N-acyl are successfully differentiated. To our surprise, some unusual structures including odd chain LCB (17:1;O2, 19:1;O2), long LCB (22:1;O2), and odd chain N-acyls (C15:1, C17:1, C27:1) exist in brain GSLs. We also reveal that the MUFA N-acyls (C22:1-C26:1) are contributed by a diverse group of C = C location isomers, including n-7, n-8, n-9, n-10, and n-12. Such a profile is different from MUFA detected in phospholipids (C16:1–C20:1), indicating the involvement of a distinct network of desaturases and elongases in the biosynthesis pathways of GSLs.

A strength of the developed method is to conduct relative compositional analysis of structural analogues or isomers at multiple levels. This capability is essential for the discovery of differential lipids from small-size clinical samples, a critical step before embarking large-scale and expensive clinical studies. Because of the lack of molecular features to differentiate IDH-wildtype from normal, we intentionally choose lipidomic phenotyping of human glioma tissue samples as an example to showcase the capability of our approach. As pointed out in Fig. 7a, compositional analysis at detailed structural levels, e.g., hFA, %C24, %SPD, and %n-9, is always associated with significantly reduced inter-sample variations as compared with the conventional profiling method where structural details are not differentiated. Consequently, 35 groups of GSLs are discovered with significant changes amongst normal, IDH-mutant,

and IDH-wild samples while only eight are found by relative quantitation at the sum composition level. The observation of decreased % hFA and %C24 in glioma tissue relative to normal provides critical clues to discover the metabolic origin for such changes. FA2H which is responsible for adding an OH group at the C2 position on fatty acids and CerS2 which prefers to assemble C24-CoA to the long chain base are both down-regulated at transcription and protein levels in glioma tissue. These data provide insights into glioma metabolism. Furthermore, we show that normal, IDH-wildtype, and IDH-mutant tissue samples can be correctly clustered using the top 20 molecular features from compositional analysis at detailed structural levels. Importantly, differences in %C24 and %hFA are key in distinguishing IDH-wildtype from normal samples, which may serve as potential molecular markers for IDH-wildtype glioma tissue. Overall, this study emphasizes the need to perform selective enrichment for lipid classes of low abundances and the importance of conducting lipidomic profiling at detailed structural levels. We believe this highly sensitive workflow for GSLs can be a routine practice for both basic and translational researchers, as GSLs are increasingly linked to metabolic and neurological diseases.

## Methods

The human brain tissue samples were provided by Huashan Hospital, Fudan University, China. Informed consent was obtained from all participants. Procedures related to these samples were compliant with ethical regulations set by the Ethical Review Board of Tsinghua University (IRB No. 20180030).

### Samples

Lipid standards and polar lipid extract from porcine brain were purchased from Avanti polar lipids, Inc. (Alabaster, AL, USA) and Cayman Chemical (Ann Arbor, Michigan, USA). Organic solvents, salts, PB reagents, and antibodies were obtained commercially. The nomenclature of GSLs, details for the synthesis of $TiO_2$ MNPs, protocols for lipid extraction, total fatty acid analysis, western blotting, and qRT-PCR are provided in the Supplementary information. The human brain tissue samples were provided by Huashan Hospital, Fudan University, China. Informed consent was obtained from all participants. Procedures related to these samples were compliant with ethical regulations set by the Ethical Review Board of Tsinghua University (IRB No. 20180030).

### Selective enrichment of GSLs

The lipids and IS (HexCer 18:1;O2/15:0) were mixed with $TiO_2$ MNPs in loading buffer (ACN/aqueous $NH_3$, 94/6, v/v, 400 μL) before subjection to 1 h agitation. $TiO_2$ MNPs were separated by magnet and washed by washing buffer (MeOH/aqueous $NH_3$, 96/4, v/v, 20 mM $NH_4HCO_3$) for two times (2×400 μL). GSLs were released by 2 h agitation in an eluting buffer (MeOH/FA, 95/5, v/v). We mixed a total of 40 μg of three phospholipid standards representative of PC, PE, and SM and five GSL standards (0.008 – 2 μg each), including HexCer 18:1;O2/15:0, HexCer 18:1;O2/18:1, HexCer 18:1;O2/24:1, HexCer 18:1;O2/18:0;2OH, and Hex2Cer 18:1;O2/17:0 with 5 mg of $TiO_2$ MNPs and 400 μL of loading buffer. For the enrichment of GSLs from porcine brain lipid extracts, 100 μg of porcine brain lipids were mixed with 5 mg of $TiO_2$ MNPs, 100 pmol of IS (HexCer 18:1;O2/15:0), and 400 μL of loading buffer. For the enrichment of GSLs from human brain lipid extracts, lipids from 400 μg of human brain tissue was mixed with 5 mg of $TiO_2$ MNPs, 200 pmol of IS (HexCer 18:1;O2/15:0), and 400 μL of loading buffer. The enrichment process followed the same as described above. Further details are provided in the Supplementary information.

### Offline PB derivatization

The PB derivatization was performed using a homemade flow microreactor. Lipid extract and 10 mM 2-acpy were dissolved in ACN/water

(5/1, v/v, 100 μL). The solution was deoxidized by nitrogen for 10 min and injected into the flow microreactor for 15 s UV irradiation (-254 nm). The reaction products were collected in a glass vial for subsequent RPLC-MS/MS. Further details are provided in the Supplementary information.

## RPLC-MS/MS

All LC separations were performed on a 20ADLC system (SHIMADZU, Kyoto, Japan) connected to an X500R QTOF mass spectrometer (SCIEX, Toronto, Canada). Intact GSLs and derivatized GSLs were analyzed using an XBridge BEH C18 Column (2.1 mm × 100 mm, 2.5 μm, Waters, Milford, MA, USA). Detailed parameters for RPLC separation and MS instrumentation are provided in Supplementary information. All the precursor windows used in each SWATH-MS/MS method are listed in Supplementary Data 7.

## Data analysis

SCIEX OS (3.0.03339) was used for data acquisition and processing. Explore was utilized for manual data evaluation, while Analytics was employed for peak integration. GSL identification was carried out manually based on RPLC retention time, accurate *m/z*, and fragmentation rules.

## Statistics and reproducibility

No statistical method was used to predetermine sample size. No data were excluded from the analyses. The experiments were not randomized. The Investigators were not blinded to allocation during experiments and outcome assessment.

## Reporting summary

Further information on research design is available in the Nature Portfolio Reporting Summary linked to this article.

# Data availability

The data generated in this study are provided in the Supplementary Information/Source Data file. The raw MS data are available from Figshare (https://doi.org/10.6084/m9.figshare.24771954). The derived MS data generated in this study have also been deposited in the MetaboLights database[48] [www.ebi.ac.uk/metabolights/MTBLS10400]. Source data are provided with this paper.

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

## Acknowledgements
Financial support from National Natural Science Foundation of China (No. 22225404 and 22227807) and National Key R&D Program of China (2018YFA0800903 and 2022YFC2406701) is greatly appreciated. The authors would like to thank Prof. Wei Hua from Huashan Hospital of Fudan University for providing clinical samples.

## Author contributions
Y.X. and Z.W. conceived the project. Z.W., J.W., performed the research; Z.W., D.Z. analyzed the data. Y.X., Z.W. and W.Z. co-wrote the paper. All authors discussed the results and commented on the paper.

## Competing interests
The authors declare no competing interests.
