## [Peer Review File · Nature Communications]

Illuminating the dark space of neutral glycosphingolipidome by selective enrichment and profiling at multi-structural levelsREVIEWER COMMENTS

Reviewer #1 (Remarks to the Author):

This manuscript reports a new methodology for the comprehensive characterization of some classes of GSL in biological samples. The combination of several approaches is used for the detailed characterization of GSL structures. The key novel aspect of this work is the improvement of the sample preparation protocol because GSL are low abundant components of the whole lipidome, and many of GSL would not be detected without this enrichment, as explained on page 10. The combination of a new enrichment strategy with TiO₂ together with RP-UHPLC separation and offline Paterno-Buchi reaction for the determination of double bond position is a smart approach for detailed characterization of HexCer, which results in the most extensive list of HexCer ever reported. In general, the manuscript is written with a smart logical flow, and it is pleasant to read it. The innovation level of this work is high.

Particular comments

1/ In my opinion, the title does not reflect well the content of this manuscript. After the first reading of the title "Illuminating the dark space of glycosphingolipidome by ...", I anticipated that the manuscript would report the analysis of many classes of GSL, including various subclasses of gangliosides, globosides, sulfatides, etc. Then I read the whole manuscript, and I have been slightly disappointed that only 3 classes of GSL (HexCer, Hex2Cer, and SHexCer) are reported, which is a discrepancy with my expectations after reading the title. I suggest revising the title to keep consistency with the content. In my opinion, the quality of this work is high enough, but the title should not be exaggerated. I think that the article title should primarily highlight the extremely detailed characterization of hexosylceramides, which has never been shown to such an extent so far.

2/ I feel a certain disbalance between the claims on the benefit of this new sample preparation protocol using TiO₂ and only 3 reported classes of GSL. I am surprised that you report an extremely high number of HexCer (214), high number of SHexCer (80), but a relatively low number of Hex2Cer (10), and the complete absence of Hex3Cer and Hex4Cer because these classes have been reported in some previous works (e.g., reference [16]) using relatively standard sample prep protocols. If the number of reported HexCer is so extremely high, then I would logically expect a higher number of Hex2Cer and the presence of some Hex3Cer and Hex4Cer. Please could you comment on the reason, why this is like this. On page 10, the authors emphasize that they detect 304 GSL after this enrichment in comparison to 133 without enrichment, which is really a significant improvement, therefore, I would expect a similar success for other Hex2Cer to Hex4Cer. Maybe you should check the data once more to be sure that you have not overlooked something. Anyway, it is important to include this information in the manuscript so the readers can also understand the limitations of the methodology.

3/ Please follow the latest shorthand nomenclature for lipid annotation. The style like HexCer d18:1/18:1 and other examples containing "d" or "t" does not reflect the latest nomenclature, which should be written in the style 18:1;O2 or O3.

4/ Page 6 – you write about 5 types of GSL standards, but I cannot find the description of these 5 standards. Please add it.

5/ Annotation NH₃.H₂O (with the bold dot in between) looks unusual for me. I think that it would be easier to report as aqueous solution of NH₃ or NH₃ in water or something like this.

6/ Page 7 – expression "...PC 34:1 and PC 36:1 are largely removed" does not provide accurate information because we do not know whether you remove 90, 99, or eventually 99.9%, which is important. Please be more specific.

7/ In Fig. 3 and elsewhere, the authors use these symbols for the annotation of carbon number and double bond number: HexCer X:omega. Personally, I would rather use the different symbol for the number of double bonds because small omega letter omega is established symbol for the assignment of the distance of double bond from the end, such as omega-3 or omega-6. This is slightly confusing for me, and I would prefer the simple form like HexCer X:Y, I do not insist on this and suggest it only for consideration of authors.

Reviewer #2 (Remarks to the Author):

This is an elegant method to isolate low abundant GSLs from biological samples. Subsequent derivatization by photochemistry not only increases sensitivity by charge-tagging, but also provides structural information about the double bonds. The combination of the methods is a valuable way to obtain a lot of information in a single measurement. The methods described in the supplementary information are solid and to be commended on.

This is an innovative and elegant methodological paper that deserves its title "Illuminating the dark space of..."

For me there are two ambiguities:

RPLC conditions for GSLs: The gradient starts highly non-polar, which seems unusual to me for RP methods. Is that correct?

Relative quantification: Figure 7 shows that the internal standard method was used for quantification. However, no addition of the internal standard is mentioned in the sample preparation (S14).

Reviewer #3 (Remarks to the Author):

The work by Z. Wang <et al.> entitled "Illuminating the dark space of glycosphingolipidome by selective enrichment and profiling at multi-structural levels" is a well-written, clear manuscript including very important data. My review concentrates on the general evaluation of the nanoparticles' synthesis and utilization, including the experimental steps for their application as enrichers for specific lipids. There are some additional general comments from my side at the end of this review report.

- The introduction lacks some kind of explanation about the focus on using nanoparticles. The authors write a lot about the significance of glycosphingolipids and about the difficulties and the importance of their quantification, however, they only report superficially about the usage of titanium oxide decorated iron oxide nanoparticles in particular. Is the specificity of the Ti(IV) for GSL based only on the authors' previous work or are there other researchers showing similar specificity for these or other types of nanoparticles? Why are nanoparticles valuable here? Perhaps the authors can add something on these points in their introduction. The authors write between lines 96 and 99 that the fact that their particles "remove phospholipids while selectively enrich GSLs in biological samples" (lines 80/81 of the main manuscript) might be a pH issue. However, you never cite the pHs of your incubation, washing, and releasing steps, as I mention again in my comments below. It is necessary to provide some clarity on this point.
- In line 101 you call your particles Fe₃O₄@TiO₂, while in the following paragraph you call them TiO₂ MNPs and in the supporting information, S-13, "Synthesis of Mag TiO₂ nanomaterials" you call them Mag TiO₂ materials. Consistency would be here helpful for clarity; I suggest standardizing this in the manuscript and the supporting information.
- Surprisingly your reference 23 (Lu, J., Wang, M., Deng, C. & Zhang, X. Facile synthesis of Fe₃O₄@mesoporous TiO₂ microspheres for selective enrichment of phosphopeptides for phosphoproteomics analysis. *Talanta* 105, 20-27 (2013)), the only source you cite on the nanoparticle synthesis and characterization topic, does not include a description of the iron oxide nanoparticle synthesis. Section 3.1 of reference 23 claims that the description of the synthesis of iron oxide is given elsewhere, but no "elsewhere" in the text of reference 23 can be found. No information is provided about the educts and the synthesis paths for the iron oxide synthesis. Therefore, it is of particular importance that you add this information accurately to the supporting information of this manuscript. Unfortunately, the current supporting information contains very little data about the nanoparticles.

Moreover, the data included are not the essential data for understanding the properties of the material.

- In the paragraph from line 110 to line 125, a lot of information is missing. Were your particles freshly prepared for the experiments reported in this manuscript or were they older particles from previous experiments? It seems they were a new charge. If not, other questions regarding storage and aging would arise. But from the size difference to the particles in reference 23, it seems that this was a newly synthesized material charge. Based on the comparison of the TEM and SEM pictures in reference 23, the new particles in the current manuscript appear to be smaller in diameter and are more homogenous in size. It is necessary to confirm the properties of your particles and characterize them to ensure that the properties are as expected from your previous experience (ref. 23, which dates from 2013). Every new charge needs at least some minimum characterization to validate its physicochemical properties: size and size distribution, surface charge, composition, etc. It would be also very helpful to have data on the saturation magnetization of your final particles to make sure that the calcination procedure did not negatively affect their magnetic moment and to have numbers on their magnetic response. This data is a common basic characteristic of magnetic nanoparticles that demonstrates that they can properly be separated using magnetic forces.
- The surface-to-volume ratio (in the form of the specific surface area) of your particulate material is of critical importance because this ratio determines the surface area available for lipid adsorption and to some extent the selectivity.
- Your particles are probably not pure Fe₃O₄ but magnetic iron oxides. If you have no demonstration of their main chemical composition, you should probably not exactly specify the species name, as the initial stoichiometry is usually not enough to identify this, and high temperatures together with environmental oxygen species usually lead to oxidized shells for both types of nanoparticles.
- Beginning with line 110 there is an explanation of the experimental procedure for incubating the nanoparticles with the lipids, but it is not clear enough. Do you statically incubate the running buffer for your HPLC analytics with the Fe₃O₄@TiO₂ nanoparticles and the lipid samples? What is the pH of the incubation system? You write in your text that the particles are negatively charged in the media, however, have you verified their charge? No information is provided on surface charge or the zeta potential of the particles in your previous publication from 2013 in Talanta, ref. 23. Can you also add the pH of the washing and the releasing buffer? The concentration of the particles during incubation, washing, and releasing is absent as is the reaction volume, which also plays a role due to interfacial effects. Please add these details to your experimental description. The reaction volumes are added later in the manuscript. However, no exact and complete data about the incubated masses neither concentrations can be found.
- For the enrichment experiments it is necessary to provide the mass ratio for the nanoparticles:lipids or nanoparticles:biological sample. This data could also not be found by this reviewer. The mass ratio defines the partitioning of the lipids on the particles' surface and is therefore key to understanding the selectivity effect. Line 559 reports how many particles were used in the incubation experiments, and the volume ratios of the solvents, but not how much lipid mass was present nor how much biological sample was added. This information needs to be added.
- In line 114 you presume that the phospholipids and the particles are negative at the incubation pH. Not only is the incubation pH lacking (as written before in this review report) but so is the explanation of why you expect your glycosphingolipids (GSL) not to be also negatively charged at this pH. Here, you must present and discuss the pH-dependent charge of natural GSL or add a suitable reference on that point.

For me as a reviewer and a scientist, after reading this impressive work there is one essential question left: Can you verify that there were no lateral reactions during your incubation experiments, for instance, catalyzed by the nanoparticles? How can you be certain that all lipid species you quantified originated from their presence in the sample and did not evolve due to reactions in the presence of a material as catalytically active as your nanoparticles?

Due to the great variety of lipid types and concentrations in the samples analyzed in this study, the second essential question in my eyes is, why does the concentration work for some types of lipids

better than for others? What is the origin of the selectivity effect? The authors do not explain the enrichment mechanism at all, even if they cite one of their previous articles where they enriched other molecules using titanium oxide nanoparticles. Even if their goal is to demonstrate that a significant concentration of difficult-to-quantify lipids is possible, as is a clear enhancement of resolution for these lipids' quantification, at least a comment on the presumed enrichment mechanism and selectivity enhancement would be very valuable, even if it is only added as a conjecture, or based on their previous experience. Enrichment can probably only happen due to interfacial properties of the stronger enriched lipids. Perhaps the authors should hazard adding a comment on this central question.

One last important point is whether the total chemical composition of the different sample types might influence the enrichment output. The different sample types, from the standard ones to the different environments and compositions of the different types of samples, can lead to different types of structural units in suspension, mainly due to the interfacial properties of some of the molecules present. This might facilitate the identification and quantification of some types of lipids compared to others among the different sample types, e.g. if some types of lipids build or are enclosed in more stable structures, there might be encapsulation barriers for the analytics which differently impact the different types of samples. Can you guarantee that your incubation/washing and releasing steps lead to accessibility to all lipids in a complete manner? Have you tested variations in sample treatments to verify that your results correspond to the real composition of the samples and are not influenced by the experimental method, i.e. by the experimental methodology chosen?

A statistical evaluation of the robustness of the data presented in the manuscript is typically necessary and lacking here. For instance, line 119 reports a recovery of 75-121%, which is also shown in figure 2d. The deviations for triplicates are brutally high. Surprisingly, the highest concentration, already in the μM range, is still clearly over 100%, which is intriguing. This result indicates that either the signals or the methodology leads to strong deviations. Did the authors also test analytical replicates? Do they know where the main error sources are and how robust their methodology is? The general evaluation of potential sources of error and the analytical reproducibility of the data should be clarified.

Point-by-point response

Response to Reviewer #1:

1. In my opinion, the title does not reflect well the content of this manuscript. After the first reading of the title “Illuminating the dark space of glycosphingolipidome by ...”, I anticipated that the manuscript would report the analysis of many classes of GSL, including various subclasses of gangliosides, globosides, sulfatides, etc. Then I read the whole manuscript, and I have been slightly disappointed that only 3 classes of GSL (HexCer, Hex2Cer, and SHexCer) are reported, which is a discrepancy with my expectations after reading the title. I suggest revising the title to keep consistency with the content. In my opinion, the quality of this work is high enough, but the title should not be exaggerated. I think that the article title should primarily highlight the extremely detailed characterization of hexosylceramides, which has never been shown to such an extent so far.

Response: We highly appreciate the constructive criticisms from the reviewer. The selective enrichment method developed in the study targets neutral GSLs according to its working principles. Therefore, we agree that the title should be revised to reflect this aspect. In the revised manuscript, we extended the discussion on the selectivity for neutral GSLs but not for acidic GSLs. We further evaluated the recovery rate for other subclasses of neutral GSLs, such as Hex3Cer and Hex4Cer (Gb4). The data showed good recovery for these subclasses although they were not detected from our analysis (please refer to response to question 2).

Changes:

The title is revised to:

Illuminating the dark space of **neutral** glycosphingolipidome by selective enrichment and profiling at multi-structural levels.

In the main text the analysis of neutral GSLs is clarified.

2. I feel a certain disbalance between the claims on the benefit of this new sample preparation protocol using TiO₂ and only 3 reported classes of GSL. I am surprised that you report an extremely high number of HexCer (214), high number of SHexCer (80), but a relatively low number of Hex2Cer (10), and the complete absence of Hex3Cer and Hex4Cer because these classes have been reported in some previous works (e.g., reference [16]) using relatively standard sample prep protocols. If the number of reported HexCer is so extremely high, then I would logically expect a higher number of Hex2Cer and the presence of some Hex3Cer and Hex4Cer. Please could you comment on the reason, why this is like this. On page 10, the authors emphasize that they detect 304 GSL after this enrichment in comparison to 133 without enrichment, which is really a significant improvement, therefore, I would expect a similar success for other Hex2Cer to Hex4Cer. Maybe you should check the data once more to be sure that you have not overlooked something. Anyway, it is important to include this information in the manuscript so the readers can also understand the limitations of the methodology.

Response: We appreciate the suggestion to check for the presence of other subclasses of GSLs, such as Hex3Cer and Hex4Cer (Gb4). After a thorough examination of the data, we did not detect any Hex3Cer and Hex4Cer (also referred to as Gb4) in either porcine brain lipids or human brain tissue samples. Our data are consistent with the sub-class distribution reported by Fuchs et al. and Fujiwara et al., where thin layer chromatography separation and chiral LC were employed for the separation, respectively, and MS for detection. Notably, the literature (Ref. [16]) mentioned by the reviewer reported the detection of Hex3Cer and Gb4 from human plasma instead of brain lipids. Given the tissue type dependence of GSLs, the relative

abundance of Hex3Cer and Gb4 in brain tissues might be below the detection limit of our method.

We further evaluated the recovery rates for three standard lipids representative of Hex3Cer, Gb4, and SHex2Cer, which showed excellent to good recovery for these three subclasses (data added as Supplementary Figure 6a). The data thus suggest the selective enrichment method does not have a bias toward Hex3Cer and Gb4.

Supplementary Fig. 6. a. %Recovery of lipid standards: Hex3Cer 18:1;O2/18:0, Gb4 18:1;O2/24:0, and SHex2Cer 18:1;O2/18:0 in a lipid mixture. The mixture contains 30 μ M of phosphatidylcholine (PC), phosphatidylethanolamine (PE), and sphingomyelin (SM), respectively, along with initial concentrations of Hex3Cer, Gb4, and SHexCer at 1,000 nM. Error bars illustrate \pm standard deviations (SD) from 3 technical replicates.

Changes: Page 8, the following sentences are added:

“We hypothesize that the C3-sulfation disrupts the cis-diol configuration and weakens the coordination of SHexCer with Ti(IV) during the loading step, leading to compromised recovery. Furthermore, the current selective enrichment condition is not optimal for gangliosides, because they can form stable carboxylic acid-Ti(IV) coordination under acidic washing condition and thus cannot be effectively eluted from the surface³³. Consequently, in later analysis we primarily target neutral GSLs from brain lipid extracts.”

[33] Palmisano, G. et al. Selective enrichment of sialic acid-containing glycopeptides using titanium dioxide chromatography with analysis by HILIC and mass spectrometry. *Nature Protocols* 5, 1974-1982 (2010).

Page 18, the following sentences are added:

“It is worth noting that HexCer and Hex2Cer are the two subclasses of neutral GSLs found in porcine brain, while complex neutral GSLs, such as Hex3Cer or Gb4, are not detected. This result, however, is consistent with previous reports utilizing thin layer chromatography separation followed by MS analysis³⁹, or chiral LC separation paired with MS analysis⁴⁰”

[39] Fuchs, B., Nimptsch, A., Süß, R. & Schiller, J. Analysis of Brain Lipids by Directly Coupled Matrix-Assisted Laser Desorption Ionization Time-of-Flight Mass Spectrometry and High-Performance Thin-Layer Chromatography. *Journal of AOAC International* 91, 1227-1236 (2008).

[40] Fujiwara, Y., Hama, K. & Yokoyama, K. Mass spectrometry in combination with a chiral column and multichannel-MRM allows comprehensive analysis of glycosphingolipid molecular species from mouse brain. *Carbohydrate Research* 490, 107959 (2020).

3. Please follow the latest shorthand nomenclature for lipid annotation. The style like HexCer d18:1/18:1 and other examples containing “d” or “t” does not reflect the latest nomenclature, which should be written in the style 18:1;O2 or O3.

Response/change: Thanks for the suggestion. We have updated the nomenclature throughout the main text and supplementary files and updated the reference.

4. Page 6 – you write about 5 types of GSL standards, but I cannot find the description of these 5 standards. Please add it.

Response/change: In page 8, the following information is added:

“Recovery was evaluated for five GSL standards (HexCer 18:1;O2/15:0, HexCer 18:1;O2/18:1, HexCer 18:1;O2/24:1, HexCer 18:1;O2/18:0;2OH and Hex2Cer 18:1;O2/17:0)...”

5. Annotation NH₃.H₂O (with the bold dot in between) looks unusual for me. I think that it would be easier to report as aqueous solution of NH₃ or NH₃ in water or something like this.

Response/change: Thanks for the suggestion. We have revised the description as aqueous NH₃.

6. Page 7 – expression “...PC 34:1 and PC 36:1 are largely removed” does not provide accurate information because we do not know whether you remove 90, 99, or eventually 99.9%, which is important. Please be more specific.

Response: Thanks for the suggestion. We evaluated the efficiency for the removal of PCs by comparing the signals of PC 36:1 ([M + H]⁺, *m/z*, 788.5 Da) and PC 34:1 ([M + H]⁺, *m/z*, 760.5 Da) in porcine brain before and after and selective enrichment. Over 99% of both PC lipids were removed. The data are added in SI and the following sentence is added.

Change: Page 9. “...PC 34:1 and PC 36:1 were removed with high efficiency (>99%, Supplementary Table 1),...”

7. In Fig. 3 and elsewhere, the authors use these symbols for the annotation of carbon number and double bond number: HexCer X:omega. Personally, I would rather use the different symbol for the number of double bonds because small omega letter omega is established symbol for the assignment of the distance of double bond from the end, such as omega-3 or omega-6. This is slightly confusing for me, and I would prefer the simple form like HexCer X:Y, I do not consist on this and suggest it only for consideration of authors.

Response/change: Thanks for the suggestion. We have replaced the symbol for the double bond number with 'Y', in the text and figures (Fig. 3i).

Response to Reviewer #2

This is an elegant method to isolate low abundant GSLs from biological samples. Subsequent derivatization by photochemistry not only increases sensitivity by charge-tagging, but also provides structural information about the double bonds. The combination of the methods is a valuable way to obtain a lot of information in a single measurement. The methods described in the supplementary information are solid and to be commended on.

This is an innovative and elegant methodological paper that deserves its title “Illuminating the dark space of...”

Response: We highly appreciate the constructive criticisms from the reviewer.

For me there are two ambiguities:

1. RPLC conditions for GSLs: The gradient starts highly non-polar, which seems unusual to me for RP methods. Is that correct?

Response: The gradient although is more non-polar than typical RPLC separations for phospholipids, however, has been utilized for glycosphingolipids. This elution gradient used in this study was optimized from Sugawara et al. (J. Oleo Sci. 2010, 59, 387), in which an isocratic elution of GSLs by acetonitrile/water (95:5, v/v) showed good performance on a C18 column.

Change: Supplementary information (S-21), the following sentence is added.

“The gradient elution was modified from an isocratic elution (acetonitrile/water = 95:5, v/v) reported by Sugawara et al.⁸”

[8] Sugawara, T., Aida, K., Duan, J. & Hirata, T. Analysis of Glucosylceramides from Various Sources by Liquid Chromatography-Ion Trap Mass Spectrometry. Journal of Oleo Science 59, 387-394 (2010).

2. Relative quantification: Figure 7 shows that the internal standard method was used for quantification. However, no addition of the internal standard is mentioned in the sample preparation (S14).

Response/change: The information of the internal standard is added in supplementary information (page S-20):

“The internal standard (HexCer 18:1;O2/15:0, 200 pmol) was added to the solution of lipid extract from 400 µg of human brain tissue.

Response to Reviewer #3

The work by Z. Wang <et al.> entitled “Illuminating the dark space of glycosphingolipidome by selective enrichment and profiling at multi-structural levels” is a well-written, clear manuscript including very important data. My review concentrates on the general evaluation of the nanoparticles’ synthesis and utilization, including the experimental steps for their application as enrichers for specific lipids. There are some additional general comments from my side at the end of this review report.

Response: We appreciate the reviewer’s constructive criticisms. We have expanded discussions on the possible mechanism for selective enrichment of GSLs, added additional data for the characterization of TiO₂ MNPs, and reported the analytical performances of the selective enrichment method. Detailed Responses are shown below.

1a. The introduction lacks some kind of explanation about the focus on using nanoparticles. The authors write a lot about the significance of glycosphingolipids and about the difficulties and the importance of their quantification, however, they only report superficially about the usage of titanium oxide decorated iron oxide nanoparticles in particular.

Change: Thanks for the suggestion. We added the importance of magnetic nanoparticles and titanium dioxide in the sample preparation step of biological samples in page 5:

“...Selective enrichment is commonly employed before MS analysis to improve detection for low abundance constituents, such as glycoproteins and phosphoproteins, in proteomics studies²⁰. As compared to micro-columns²¹, dispersed microspheres²², or pipette tips²³, functional magnetic nanoparticles exhibit attractive features for selective enrichment due to their large surface area, flexible surface modification capabilities, and strong magnetic responsiveness^{24, 25}.”

[20] Hughes, C.S. et al. Single-pot, solid-phase-enhanced sample preparation for proteomics experiments. *Nature Protocols* 14, 68-85 (2019).

[21] Larsen, M.R., Thingholm, T.E., Jensen, O.N., Roepstorff, P. & Jørgensen, T.J.D. Highly Selective Enrichment of Phosphorylated Peptides from Peptide Mixtures Using Titanium Dioxide Microcolumns. *Molecular & Cellular Proteomics* 4, 873-886 (2005).

[22] Leitner, A. et al. Probing the Phosphoproteome of HeLa Cells Using Nanocast Metal Oxide Microspheres for Phosphopeptide Enrichment. *Analytical Chemistry* 82, 2726-2733 (2010).

[23] Huang, P., Li, H., Gao, W., Cai, Z. & Tian, R. A Fully Integrated Spintip-Based Approach for Sensitive and Quantitative Profiling of Region-Resolved in Vivo Brain Glycoproteome. *Analytical Chemistry* 91, 9181-9189 (2019).

[24] Deng, H. et al. Monodisperse Magnetic Single-Crystal Ferrite Microspheres. *Angewandte Chemie International Edition* 44, 2782-2785 (2005).

[25] Li, Y., Zhang, X., Deng, C. Functionalized magnetic nanoparticles for sample preparation in proteomics and peptidomics analysis. *Chemical Society Reviews* 42, 8517-8539 (2013).

1b. Is the specificity of the Ti(IV) for GSL based only on the authors’ previous work or are there other researchers showing similar specificity for these or other types of nanoparticles? Why are nanoparticles valuable here? Perhaps the authors can add something on these points in their introduction.

Response/Change: Ti(IV)-cis-diol coordination is well studied and has been used in sample preparation to enhance separations of glycosphingolipids, glycopeptides, and nucleoside from complex matrices. The information and associated references are added in page 6.

“Ti (IV) in TiO₂ exhibits strong coordination with cis diol across a broad pH range (1.5-14)²⁵. This chemistry has been harnessed for separations of glycosphingolipids^{26, 27}, glycopeptides²⁸, and nucleosides²⁹ from complex matrices.”

[25] Wang, S.-T., Huang, W., Lu, W., Yuan, B.-F. & Feng, Y.-Q. TiO₂-Based Solid Phase Extraction Strategy for Highly Effective Elimination of Normal Ribonucleosides before Detection of 2'-Deoxynucleosides/Low-Abundance 2'-O-Modified Ribonucleosides. *Analytical Chemistry* 85, 10512-10518 (2013).

[26] Huang, Z., Wu, Q., Lu, H., Wang, Y. & Zhang, Z. Separation of Glycolipids/Sphingolipids from Glycerophospholipids on TiO₂ Coating in Aprotic Solvent for Rapid Comprehensive Lipidomic Analysis with Liquid Microjunction Surface Sampling-Mass Spectrometry. *Analytical Chemistry* 92, 11250-11259 (2020).

[27] Noda, A., Kato, M., Miyazaki, S. & Kyogashima, M. Separation of glycosphingolipids with titanium dioxide. *Glycoconjugate Journal* 35, 493-498 (2018).

[28] Wang, S.-T. et al. “Old” metal oxide affinity chromatography as “novel” strategy for specific capture of cis-diol-containing compounds. *Journal of Chromatography A* 1361, 100-107 (2014).

[29] Wu, Q., Wu, D. & Guan, Y. Hybrid Titania-Zirconia Nanoparticles Coated Adsorbent for Highly Selective Capture of Nucleosides from Human Urine in Physiological Condition. *Analytical Chemistry* 86, 10122-10130 (2014).

1c. The authors write between lines 96 and 99 that the fact that their particles “remove phospholipids while selectively enrich GSLs in biological samples” (lines 80/81 of the main manuscript) might be a pH issue. However, you never cite the pHs of your incubation, washing, and releasing steps, as I mention again in my comments below. It is necessary to provide some clarity on this point.

Change: Thanks for the suggestion. The pH values for the loading, washing, and eluting buffer solutions are 11.83, 10.18 and 1.29, respectively. The information is added in page 7 of the main text. The possible mechanism is discussed. Please see response to Question 11.

2. In line 101 you call your particles Fe₃O₄@TiO₂, while in the following paragraph you call them TiO₂ MNPs and in the supporting information, S-13, “Synthesis of Mag TiO₂ nanomaterials” you call them Mag TiO₂ materials. Consistency would be here helpful for clarity; I suggest standardizing this in the manuscript and the supporting information.

Response/change: We have unified the name as TiO₂ MNPs throughout the main text and Supplementary files.

3a. Surprisingly your reference 23 (Lu, J., Wang, M., Deng, C. & Zhang, X. Facile synthesis of Fe₃O₄@mesoporous TiO₂ microspheres for selective enrichment of phosphopeptides for phosphoproteomics analysis. *Talanta* 105, 20-27 (2013)), the only source you cite on the nanoparticle synthesis and characterization topic, does not include a description of the iron oxide nanoparticle synthesis. Section 3.1 of reference 23 claims that the description of the synthesis of iron oxide is given elsewhere, but no “elsewhere” in the text of reference 23 can be found. No information is provided about the educts and the synthesis paths for the iron oxide synthesis. Therefore, it is of particular importance that you add this information accurately to the supporting information of this manuscript.

Response: We have added the correct reference, including the synthesis procedure for Fe₃O₄ nanoparticles in the main text (Page 6, ref 29) and the supporting information (page S-19, ref 2).

[29] Qi, D., Lu, J., Deng, C., Zhang, X. Magnetically Responsive Fe₃O₄@C@SnO₂ Core–Shell Microspheres: Synthesis, Characterization and Application in Phosphoproteomics. *The Journal of Physical Chemistry C* 113, 15854-15861 (2009).

3b. Unfortunately, the current supporting information contains very little data about the nanoparticles. Moreover, the data included are not the essential data for understanding the properties of the material.

Response/change: We conducted a series of characterization of the nanoparticles, including the zeta potential, composition, saturation magnetization, and surface-to-volume ratio of TiO₂ MNPs. These data are added as Fig 1,2 in the Supplementary Information. Please see detailed response to Questions 4 and 5.

4a. In the paragraph from line 110 to line 125, a lot of information is missing. Were your particles freshly prepared for the experiments reported in this manuscript or were they older particles from previous experiments? It seems they were a new charge. If not, other questions regarding storage and aging would arise. But from the size difference to the particles in reference 23, it seems that this was a newly synthesized material charge.

Response: The TiO₂ MNPs were prepared in house and not borrowed from other labs. Each batch of TiO₂ MNPs was prepared in bulk (~1 g) and then used for experiments spanning for about three months. Several batches of nanoparticles were synthesized and used during the whole project. The potential variations in selective enrichment for neutral GSLs were evaluated for TiO₂ MNPs synthesized from different batches and after 3-month storage. The data showed that phospholipids were all efficiently removed and the ion signals of the major GSLs were quite comparable for these different conditions.

Change: Page 9, the following sentences are added:

“TiO₂ MNPs synthesized from different batches exhibited less than 5% variations regarding the ion abundances of two main components of brain GSLs, HexCer 42:2;O₂ and HexCer 42:2;O₃ (Supplementary Fig. 9a-b). Moreover, the variations were less than 17% after a 3-month storage (Supplementary Fig. 9b-c, Supplementary Table 1).”

The data are added in Supplementary Figure 9 and Supplementary Table 1.

Supplementary Figure 9. Reproducibility of TiO₂ MNP-based enrichment for porcine brain GSLs using two batches of TiO₂ MNPs and aged TiO₂ MNPs. Lipid profile of 2.5 μg of porcine brain lipids per injection analyzed by RPLC-MS (RT: 5–30 min) after selective enrichment using (a-b) TiO₂ MNPs from two synthetic batches and (c) TiO₂ MNPs aged for three months.

4b. Based on the comparison of the TEM and SEM pictures in reference 23, the new particles in the current manuscript appear to be smaller in diameter and are more homogenous in size. It is necessary to confirm the properties of your particles and characterize them to ensure that the properties are as expected from your previous experience (ref. 23, which dates from 2013).

Response/change: We utilized "Nano Measure software" to calculate the size of the magnetic core and TiO₂ layer in the TiO₂ MNPs from TEM images provided in supplementary Figure 1f. The size distribution of TiO₂ MNPs was calculated according to SEM image. The new information is added in the main text and supplementary information.

Page 6

“The ferrite core was 250 nm in diameter and covered by a mesoporous TiO₂ layer of 40 nm thickness (Supplementary Fig. 1d-f)”

Supplementary Fig. 1 f. Transmission electron microscopy (TEM) image of TiO₂ magnetic nanoparticles after calcination.

Supplementary Fig. 1 b. Scanning Electron Microscope (SEM) image of TiO₂ MNPs. c. Size distribution of 105 TiO₂ MNPs, analyzed from the image in Figure S2b.

4c. Every new charge needs at least some minimum characterization to validate its physicochemical properties: size and size distribution, surface charge, composition, etc. It would be also very helpful to have data on the saturation magnetization of your final particles to make sure that the calcination procedure did not negatively affect their magnetic moment and to have numbers on their magnetic response. This data is a common basic characteristic of magnetic nanoparticles that demonstrates that they can properly be separated using magnetic forces.

Response/change: We have characterized the magnetic response, surface charge, composition of TiO₂ MNPs. Data are added in Supplementary Figure 1 h, i and Supplementary Figure 2c, d.

Page 6, the following sentences are revised.

“On average, the particles were 305 ± 22 nm in diameter, measured from 105 TiO₂ MNPs in a scanning electron microscopy (SEM) image (Supplementary Fig. 1b, c). ... Energy dispersive X-ray (EDX) and X-ray photoelectron spectroscopy (XPS) measurements of the TiO₂ MNPs proved a composite of iron oxide and titanium (IV) dioxide (Supplementary Fig. 1g-i). The TiO₂ MNPs had an average surface-to-volume ratio of 89.9 m²/g, pore volume of 0.11 cm³/g, surface charge of -0.493mV in water, and good magnetic responsiveness of 44.6 emu/g (Supplementary Fig. 2).”

Supplementary Fig. 1 h. Full scan X-ray photoelectron (XPS) spectrum of TiO₂ MNPs and (i) Ti2p XPS spectrum of TiO₂ MNPs.

Note 1: The two peaks centered at 463.85 and 458.15 eV in Fig 1i correspond to Ti2p_{1/2} and Ti2p_{3/2} binding energies. The splitting energy between them is 5.7 eV, indicating a normal state of Ti⁴⁺ in the TiO₂ MNPs.

5. The surface-to-volume ratio (in the form of the specific surface area) of your particulate material is of critical importance because this ratio determines the surface area available for lipid adsorption and to some extent the selectivity.

Response/Change: The surface-to-volume ratio of the TiO₂ MNPs was 89.9 m²/g. The information is added to the main text (Page 7) and supplementary information.

Supplementary Figure 2 a. N₂ sorption-desorption isotherms. b. Pore size distribution analysis obtained using the Barrett-Joyner-Halenda (BJH) method.

6. Your particles are probably not pure Fe₃O₄ but magnetic iron oxides. If you have no demonstration of their main chemical composition, you should probably not exactly specify the species name, as the initial stoichiometry is usually not enough to identify this, and high temperatures together with environmental oxygen species usually lead to oxidized shells for both types of nanoparticles.

Response/change: Thanks for the suggestion. We have revised the description as magnetic iron oxides.

7a. Beginning with line 110 there is an explanation of the experimental procedure for incubating the nanoparticles with the lipids, but it is not clear enough. Do you statically incubate the running buffer for your HPLC analytics with the Fe₃O₄@TiO₂ nanoparticles and the lipid samples? What is the pH of the incubation system?... Can you also add the pH of the washing and the releasing buffer?

Response: We incubated the lipid samples and nanoparticles with a loading buffer containing 94% acetonitrile and 6% ammonia. The measured pH value is 11.83. The pH values for the washing and eluting buffer solutions are 10.18 and 1.29, respectively. The information is added to the main text.

7b. You write in your text that the particles are negatively charged in the media, however, have you verified their charge? No information is provided on surface charge or the zeta potential of the particles in your previous publication from 2013 in Talanta, ref. 23.

Response: We have measured the zeta potential of the particles in water, and in a solution containing 96% methanol and 4% ammonia, with or without an addition of 20 mM NH₄HCO₃. The zeta potential of TiO₂ is -0.083 mV in the buffer containing 96% methanol and 4% aqueous ammonia, indicating an almost neutral surface. After adding NH₄HCO₃, the zeta potential shifted to +43.3 mV, proving a positively charged surface. We hypothesize that this is because TiO₂ MNPs attracted extra NH₄⁺ ions onto the surface.

Changes: The data are added in Supplementary Fig. 2c.

Supplementary Fig. 2 c. Zeta potentials of TiO₂ MNPs in different buffer solutions: water, aqueous NH₃/MeOH (4:96, v/v), aqueous NH₃/MeOH (4:96, v/v) containing 20 mM NH₄HCO₃. d. Magnetic hysteresis curves of TiO₂ MNPs.

Change: Page 7, the following sentence is added.

“...Indeed, a zeta potential of 43.3 mV was measured for TiO₂ MNPs in the ammonium containing washing buffer (Supplementary Fig. 2c).”

8c. The concentration of the particles during incubation, washing, and releasing is absent as is the reaction volume, which also plays a role due to interfacial effects. Please add these details

to your experimental description. The reaction volumes are added later in the manuscript. However, no exact and complete data about the incubated masses neither concentrations can be found.

Response/Change: The concentration of the TiO₂ MNPs and the volume of the solutions are added. The other experimental details on sample usage are added as well in the main text and supporting information.

8. For the enrichment experiments it is necessary to provide the mass ratio for the nanoparticles:lipids or nanoparticles:biological sample. This data could also not be found by this reviewer. The mass ratio defines the partitioning of the lipids on the particles' surface and is therefore key to understanding the selectivity effect. Line 559 reports how many particles were used in the incubation experiments, and the volume ratios of the solvents, but not how much lipid mass was present nor how much biological sample was added. This information needs to be added.

Response/Change: Thanks for the suggestion. Page 31 Online Methods, the following information is added:

“We mixed a total of 40 µg of three phospholipid standards representative of PC, PE, and SM and five GSL standards (0.008 – 2 µg each), including HexCer 18:1;O2/15:0, HexCer 18:1;O2/18:1, HexCer 18:1;O2/24:1, HexCer 18:1;O2/18:0;2OH, and Hex2Cer 18:1;O2/17:0 with 5 mg of TiO₂ MNPs and 400 µL of loading buffer. For the enrichment of GSLs from porcine brain lipid extracts, 100 µg of porcine brain lipids were mixed with 5 mg of TiO₂ MNPs, 100 pmol of IS (HexCer 18:1;O2/15:0), and 400 µL of loading buffer. For the enrichment of GSLs from human brain lipid extracts, lipids from 400 µg of human brain tissue was mixed with 5 mg of TiO₂ MNPs, 200 pmol of IS (HexCer 18:1;O2/15:0), and 400 µL of loading buffer. The enrichment process followed the same as described above.”

9. In line 114 you presume that the phospholipids and the particles are negative at the incubation pH. Not only is the incubation pH lacking (as written before in this review report) but so is the explanation of why you expect your glycosphingolipids (GSL) not to be also negatively charged at this pH. Here, you must present and discuss the pH-dependent charge of natural GSL or add a suitable reference on that point.

Response: The pH of the washing buffer is 10.18. The pKa values of GlcCer 18:1;O2/22:0 and GalCer 18:1;O2/24:1 are both 12.18 as listed in the human metabolome database (HMDB) (ref 31). Therefore, neutral GSLs should stay in neutral state under a pH value of 10.18.

Change: In page 7, the following information is added:

“Meanwhile, neutral GSL molecules, having the pKa slightly above 12³¹, should stay in a neutral state and retain strong coordination with TiO₂ MNPs.”

[31] Wishart, D.S. et al. HMDB 5.0: the Human Metabolome Database for 2022. *Nucleic Acids Research* **50**, D622-D631 (2022).

10. For me as a reviewer and a scientist, after reading this impressive work there is one essential question left: Can you verify that there were no lateral reactions during your incubation experiments, for instance, catalyzed by the nanoparticles? How can you be certain that all lipid

species you quantified originated from their presence in the sample and did not evolve due to reactions in the presence of a material as catalytically active as your nanoparticles?

Response/change: Thanks for the suggestion. We didn't observe side reactions during the selective enrichment. We tested the recovery of GSLs from standard lipid mixtures, with %recovery ranging from 75% to 105% for GSLs. The good recovery suggests that there hasn't been a significant loss of GSLs. Moreover, we did not observe new peaks related to GSLs or phospholipids in the separated and enriched standard samples in the MS data.

In page 9, the following sentence is added:

“Although Ti(IV) can catalyze the hydrolysis of phosphate diesters³⁷, we did not observe any hydrolysis products of GSLs or phospholipids.”

11. Due to the great variety of lipid types and concentrations in the samples analyzed in this study, the second essential question in my eyes is, why does the concentration work for some types of lipids better than for others? What is the origin of the selectivity effect? The authors do not explain the enrichment mechanism at all, even if they cite one of their previous articles where they enriched other molecules using titanium oxide nanoparticles. Even if their goal is to demonstrate that a significant concentration of difficult-to-quantify lipids is possible, as is a clear enhancement of resolution for these lipids' quantification, at least a comment on the presumed enrichment mechanism and selectivity enhancement would be very valuable, even if it is only added as a conjecture, or based on their previous experience. Enrichment can probably only happen due to interfacial properties of the stronger enriched lipids. Perhaps the authors should hazard adding a comment on this central question.

Response: The selectivity for GSL derives from a relatively strong coordination between Ti(IV) in TiO₂ and the cis-diol function group across a broad pH range (1.5-14). However, the coordination between the phospholipid and TiO₂ is more sensitive to pH, which can be largely disrupted under basic pH conditions. This is the reason a basic washing buffer solution is necessary. We also found that an addition of NH₄⁺ ions improved the efficiency to remove phospholipids. We hypothesize the extra NH₄⁺ ions can form electrostatically interactions with negative charged phosphate and further weaken its interaction with the TiO₂ MNP surface.

Changes:

In page 7, we added the following sentences to illustrate the NH₄⁺ based selective separation mechanism:

“The presence of ammonium ions in the washing buffer was found critical to remove phospholipids, which increased the washing efficiency by 3-8 times relative to the buffer without NH₄HCO₃ (4% aqueous NH₃/96% MeOH, v/v, Supplementary Fig. 3). We hypothesize that the excess of ammonium cations weakens electrostatic interactions between phospholipids and TiO₂ MNPs under basic pH condition (pH = 10.18) and separates them by forming a positive ion layer on the TiO₂ MNPs (Fig. 2a). Indeed, a zeta potential of 43.3 mV was measured for TiO₂ MNPs in the ammonium containing washing buffer (Supplementary Fig. 2c). Meanwhile, neutral GSL molecules, having the pK_a slightly above 12³¹, should stay in a neutral state and retain strong coordination with TiO₂ MNPs.”

The data illustrating the effect of NH₄⁺ are added in Supplementary Figure 3.

Supplementary Figure 3. Optimization of washing buffer to remove phospholipids. The mixture of standard lipids were prepared with a combination of PE 16:0/18:1, PE 16:0/18:1, SM 18:1;O₂/12:0, and HexCer 18:1;O₂/18:1, with a ratio of 5:5:5:1. The bars illustrate the LC peak area of [M + H]⁺ ions of each lipid in the washing buffer with or without 20 mM NH₄HCO₃. The increased abundances of phospholipids in different washing buffer solutions are indicated above the bars. Source data are provided in a Source Data file.

12a. One last important point is whether the total chemical composition of the different sample types might influence the enrichment output. The different sample types, from the standard ones to the different environments and compositions of the different types of samples, can lead to different types of structural units in suspension, mainly due to the interfacial properties of some of the molecules present. This might facilitate the identification and quantification of some types of lipids compared to others among the different sample types, e.g. if some types of lipids build or are enclosed in more stable structures, there might be encapsulation barriers for the analytics which differently impact the different types of samples.

Response: The reviewer is concerned about the impact of matrix effect on the performance of selective enrichment. We thus examined the ion abundance of HexCer d18:1/15:0 (200 pmol) which was separately added into 22 different human brain tissue samples as the internal standard. The relative standard deviation of its ion abundance was within 20%, suggesting the enrichment method is not significantly affected by matrix.

Changes:

In page 21, the following information is added:

“The RSD of IS (HexCer 18:1;O₂/15:0, 200 pmol) from these samples was less than 20%, suggesting a relatively stable performance from selective enrichment (Supplementary Fig. 18).”

The data are added in Supplementary Fig. S18

Supplementary Figure 18. RSDs of the ion abundance of IS (HexCer 18:1;O2/15:0, 200 pmol /400 µg human brain) in 22 human brain samples after selective enrichment. Source data are provided in the Source Data file.

12b. Can you guarantee that your incubation/washing and releasing steps lead to accessibility to all lipids in a complete manner? Have you tested variations in sample treatments to verify that your results correspond to the real composition of the samples and are not influenced by the experimental method, i.e. by the experimental methodology chosen?

Response: We evaluated the ion abundances of 22 neutral GSL species consisting of different chain lengths and OH groups in porcine brain before and after selective enrichment. These GSLs exhibited almost identical profiles before and after the selective enrichment (RSD <15%). The data suggest that the enrichment method would not affect the real composition of the GSLs in brain tissue samples.

Change: In page 9, the following information is added:

“We compared the relative abundances of 22 neutral GSLs consisting of different chain lengths and OH groups in porcine brain before and after selective enrichment. Almost identical profiles were obtained (RSD <15%, Supplementary Fig. 7).”

The data are added in Supplementary Information

Supplementary Figure 7. Comparisons of the ion abundances of 22 neutral GSLs in porcine brain with and without selective enrichment. The 22 neutral GSLs were selected from the RPLC retention time from 15 min -30 min to minimize interference from phospholipids. Each GSL is presented as a percentage relative to the most abundant species, HexCer 42:2;O₂, set at 100%. The dots represent RSDs of ion abundances of the 22 GSL species with and without selective enrichment. Source data are provided in the Source Data file.

13a. A statistical evaluation of the robustness of the data presented in the manuscript is typically necessary and lacking here. For instance, line 119 reports a recovery of 75-121%, which is also shown in figure 2d. The deviations for triplicates are brutally high. Surprisingly, the highest concentration, already in the μM range, is still clearly over 100%, which is intriguing. This result indicates that either the signals or the methodology leads to strong deviations.

Response: We carefully checked the MS data and found out that the ion signals of the neutral GSL were typically contributed by three forms of ions: $[\text{M} + \text{H}]^+$, $[\text{M} + \text{Na}]^+$, and $[\text{M} - \text{H}_2\text{O} + \text{H}]^+$. The %recovery shown in Fig. 2d were calculated only using $[\text{M} + \text{H}]^+$, which showed relatively large variations and concentration dependence. We reassessed the recovery rates by considering the combined intensities of $[\text{M} + \text{H}]^+$, $[\text{M} + \text{Na}]^+$, and $[\text{M} - \text{H}_2\text{O} + \text{H}]^+$ ions for each sample. The %recovery are now in the range of 75-105%. The RSDs are also reduced to 5-20%. Additionally, the larger deviations are typically associated with low concentrations. The RSDs for %recovery values are 6-20% (5 nM), 5-13% (12.5 nM), 5-8% (1250 nM). The detection limit of the current method is 20 nM (initial concentration before enrichment).

Changes:

Figure 2d is revised. In page 8, the discussion is revised as follows:

“The GSL standards were prepared at three initial concentrations (5 nM, 12.5 nM, and 1250 nM), then selectively enriched using TiO₂ MNPs, and concentrated four times before MS analysis. %Recovery was assessed by comparing the ion abundance of each GSL (sum of [M

+ H]⁺, [M + Na]⁺, and [M-H₂O+H]⁺) before and after selective enrichment. Good %recovery was achieved, ranging from 75-105% (Fig. 2d and Supplementary Fig. 4). The detection limit for the neutral GSL was estimated to be 20 nM. ”

The data are added in Supplementary Fig. 4.

13b. Did the authors also test analytical replicates? Do they know where the main error sources are and how robust their methodology is?

Response: We conducted five analytical replicates for porcine brain GSLs. The RSDs were less than 20% for five GSL species. Note that the relative abundances of these five GSLs span across three orders of magnitudes. The data thus suggest that the enrichment method provides adequate reproducibility. The data are included in supplementary Fig. 8. The comments are provided in page 9.

Change: “TiO₂ MNP-based enrichment for GSL also provided good reproducibility **in parallel experiments**. The ion signals of representative GSLs of different subclasses all exhibited less than 20% RSD from five replicates, even though their MS¹ signals spanned three orders of magnitude (Supplementary Fig. 8).”

REVIEWERS' COMMENTS

Reviewer #1 (Remarks to the Author):

I have read the response to reviews and checked changes in the manuscript. It is my pleasure to conclude that the authors have seriously considered all my comments and performed the targeted responses. Their comments and changes are accurate and fair. Concerning the change of the title based on my comment, I had initially the same idea to add only the word "neutral" as the authors did, but finally, I decided not to write it in my report. Therefore, I agree that this is the best solution. I also appreciate the additional discussion on the anticipated mechanism of this enrichment with the explanation of why this procedure does not work well for gangliosides and moderately well for sulfatides. The reported number of 80 sulfatides is high.

I am convinced that this paper will be a valuable contribution for Nature Communications, and I do not have further questions.

Reviewer #2 (Remarks to the Author):

My comments have been answered in a satisfactory manner

Reviewer #3 (Remarks to the Author):

The authors have responded carefully and in a very complete manner to all the points raised by this reviewer. In my opinion, the manuscript can be accepted in its current form. It is a thorough and highly important work that deserves to be published soon. It reports on a meaningful, reliable, and sound method to enrich glycosphingolipids and leads to a clear advance in their quantification methodology. Moreover, the article offers a large quantity of data on glycosphingolipids from brain tissue. It follows current developments in proteomics analysis, extending them to the lipidomics field. The revision carried out by the authors is detailed enough to enable experimental reproducibility.

Please, find below only very small minor revision points:

- Supplementary Figure 1b, the size bar in the image is too small to be readable. Please, add at least its value as a comment in the figure caption. In 1c value 305 ± 22 nm: the " \pm " is missing.
- Please, provide the pH at which the surface charge was measured, particularly for the zeta potential values in the manuscript as well as for Supplementary Figure 2c. Nanoparticle suspension zeta potential values are strongly pH dependent; therefore, the pH value is essential for zeta potential data. If you did not measure it, at least add to the water the information if it was neutral deionized water or distilled and deionized water (for the values in the main manuscript) and leave the rest as it is. You can also just measure the pH of the solutions and add this value in Supplementary Figure 2c, but state clearly that you are providing the pH of the solution and not of the suspension. In the presence of nanoparticles, the pH of the suspension often demonstrates a clear shift depending on the nanoparticle-to-solution-mass-to-volume ratio.
- Why is the pKa reported in the form of a power value, which seems strange? Is there a reason for this form? Usually, pKa is reported as a normal pH value. Please, change it accordingly if there is no particular reason for choosing this format: "...having the pKa slightly above 12^{31} ...".

Point-by-point response to reviewers' comments

Reviewer #1 (Remarks to the Author):

I have read the response to reviews and checked changes in the manuscript. It is my pleasure to conclude that the authors have seriously considered all my comments and performed the targeted responses. Their comments and changes are accurate and fair. Concerning the change of the title based on my comment, I had initially the same idea to add only the word "neutral" as the authors did, but finally, I decided not to write it in my report. Therefore, I agree that this is the best solution. I also appreciate the additional discussion on the anticipated mechanism of this enrichment with the explanation of why this procedure does not work well for gangliosides and moderately well for sulfatides. The reported number of 80 sulfatides is high.

I am convinced that this paper will be a valuable contribution for Nature Communications, and I do not have further questions.

Response: We appreciate the comments from this reviewer. No further revision is requested.

Reviewer #2 (Remarks to the Author):

My comments have been answered in a satisfactory manner.

Response: We appreciate the comments from this reviewer. No further revision is requested.

Reviewer #3 (Remarks to the Author):

The authors have responded carefully and in a very complete manner to all the points raised by this reviewer. In my opinion, the manuscript can be accepted in its current form. It is a thorough and highly important work that deserves to be published soon. It reports on a meaningful, reliable, and sound method to enrich glycosphingolipids and leads to a clear advance in their quantification methodology. Moreover, the article offers a large quantity of data on glycosphingolipids from brain tissue. It follows current developments in proteomics analysis, extending them to the lipidomics field. The revision carried out by the authors is detailed enough to enable experimental reproducibility.

Response: We appreciate the comments from this reviewer.

Please, find below only very small minor revision points:

1. Supplementary Figure 1b, the size bar in the image is too small to be readable. Please, add at least its value as a comment in the figure caption. In 1c value 305 ± 22 nm: the "±" is missing.

Response/change: We have increased the font size for "2 μm" of the size bar in Supplementary Figure 1b. We have added the "±" in Supplementary Figure 1c.

2. Please, provide the pH at which the surface charge was measured, particularly for the zeta potential values in the manuscript as well as for Supplementary Figure 2c. Nanoparticle suspension zeta potential values are strongly pH dependent; therefore, the pH value is essential for zeta potential data. If you did not measure it, at least add to the water the information if it was neutral deionized water or distilled and deionized water (for the values in the main

manuscript) and leave the rest as it is. You can also just measure the pH of the solutions and add this value in Supplementary Figure 2c, but state clearly that you are providing the pH of the solution and not of the suspension. In the presence of nanoparticles, the pH of the suspension often demonstrates a clear shift depending on the nanoparticle-to-solution-mass-to-volume ratio.

Response/change: Thanks for your advice. We defined “water” as “neutral distilled water” in Supplementary Figure 2c and the pH values are added.

The caption of supplementary figure 2c is revised as follows:

“c. Zeta potentials of TiO₂ MNPs in three buffer solutions: neutral distilled water, aqueous NH₃/MeOH (4:96, v/v) (pH =11.33), aqueous NH₃/MeOH (4:96, v/v) containing 20 mM NH₄HCO₃ (pH =10.18).”

In the main text, page 6, the sentence has been changed as follows:

“...pore volume of 0.11 cm³ g⁻¹, surface charge of -0.493 mV in neutral distilled water, ...”

3. Why is the pKa reported in the form of a power value, which seems strange? Is there a reason for this form? Usually, pKa is reported as a normal pH value. Please, change it accordingly if there is no particular reason for choosing this format: “...having the pKa slightly above 12³¹...”.

Response: Sorry for the confusion. The superscript 31 is the reference number for the pKa value instead of the power.